# Adaptive Compression for Communication-Efficient Distributed Training

**Maksim Makarenko**                                    *maksim.makarenko@kaust.edu.sa*
*King Abdullah University of Science and Technology*
**Elnur Gasanov**                                        *elnur.gasanov@kaust.edu.sa*
*King Abdullah University of Science and Technology*
**Rustem Islamov**                                       *rustem.islamov@ip-paris.fr*
*Institut Polytechnique de Paris*
**Abdurakhmon Sadiev**                              *abdurakhmon.sadiev@kaust.edu.sa*
*King Abdullah University of Science and Technology*
**Peter Richtárik**                                      *peter.richtarik@kaust.edu.sa*
*King Abdullah University of Science and Technology*

**Reviewed on OpenReview:** *https://openreview.net/forum?id=Rb6VDOHebB*

## Abstract

We propose Adaptive Compressed Gradient Descent (AdaCGD) – a novel optimization algorithm for communication-efficient training of supervised machine learning models with adaptive compression level. Our approach is inspired by the recently proposed three point compressor (3PC) framework of Richtárik et al. (2022), which includes error feedback (EF21), lazily aggregated gradient (LAG), and their combination as special cases, and offers the current state-of-the-art rates for these methods under weak assumptions. While the above mechanisms offer a fixed compression level or adapt between two extreme compression levels, we propose a much finer adaptation. In particular, we allow users to choose between selected contractive compression mechanisms, such as Top-$K$ sparsification with a user-defined selection of sparsification levels $K$, or quantization with a user-defined selection of quantization levels, or their combination. AdaCGD chooses the appropriate compressor and compression level adaptively during the optimization process. Besides i) proposing a theoretically-grounded multi-adaptive communication compression mechanism, we further ii) extend the 3PC framework to bidirectional compression, *i.e.*, allow the server to compress as well, and iii) provide sharp convergence bounds in the strongly convex, convex, and nonconvex settings. The convex regime results are new even for several key special cases of our general mechanism, including 3PC and EF21. In all regimes, our rates are superior compared to all existing adaptive compression methods.

## 1 Introduction

Training machine learning models is computationally expensive and time-consuming. In recent years, researchers have tended to use increasing datasets, often distributed over several *devices*, throughout the training process in order to improve the effective generalization ability of contemporary machine learning frameworks(Vaswani et al., 2019). By the word "device" or "node," we refer to any gadget that can store data, compute loss values and gradients (or stochastic gradients), and communicate with other different devices. For example, this distributed setting appears in *federated learning* (FL) (Konečnỳ et al., 2016; McMahan et al., 2017; Lin et al., 2018). FL is a machine learning setting where a substantial number of strongly heterogeneous clients attempt to cooperatively train a model using the varied data stored on these devices without violating clients' information privacy(Kairouz et al.). In this setting, distributed methods are very efficient(Goyal et al., 2017; You et al., 2019), and therefore, federated frameworks have attracted tremendous attention in recent years.

---

**Algorithm 1** DCGD method with master compression

---

1: **Input:** starting point $x^0 \in \mathbb{R}^d$; $\tilde{g}^0, g_i^0 \in \mathbb{R}^d$ for $i = 1, \cdots, n$ (known by nodes), $\tilde{g}^0 = \frac{1}{n}\sum_{i=1}^{n} g_i^0$ (known by master); learning rate $\gamma > 0$, worker compressor $\mathcal{M}^W$, master compressor $\mathcal{M}^M$.
2: **for** $t = 0,1,2,\cdots,T-1$ **do**
3:     Server broadcasts $\tilde{g}^t$ to all workers
4:     **for all devices** $i = 1, \ldots, n$ **in parallel do**
5:         $x^{t+1} = x^t - \gamma\tilde{g}^t$
6:         $g_i^{t+1} = \mathcal{M}_i^{W,t}(\nabla f_i(x^{t+1}))$
7:         Communicate $g_i^{t+1}$ to the server
8:     **end for**
9:     Server aggregates received gradient estimators $g^{t+1} = \frac{1}{n}\sum_{i=1}^{n} g_i^{t+1}$
10:    $\tilde{g}^{t+1} = \mathcal{M}^{M,t}(g^{t+1})$
11: **end for**

---

Dealing with the distributed environment, we consider the following optimization problem

$$\min_{x \in \mathbb{R}^d} \left\{ f(x) := \frac{1}{n}\sum_{i=1}^{n} f_i(x) \right\}, \tag{1}$$

where $x \in \mathbb{R}^d$ is the parameter vector of a training model, $d$ is the dimensionality of the problem (number of trainable features), $n$ is the number of workers/devices/nodes, and $f_i(x)$ is the loss incurred by model $x$ on data stored on worker $i$. The loss function $f_i : \mathbb{R}^d \to \mathbb{R}$ often has the form of expectation of some random function $f_i(x) := \mathbb{E}_{\xi \sim \mathcal{D}_i}[f_\xi(x)]$ with $\mathcal{D}_i$ being the distribution of training data owned by worker $i$. In federated learning, these distributions can be different (so-called *heterogeneous* case). This finite sum function form allows us to capture the distributed nature of the problem in a very efficient way.

The most effective models are frequently over-parameterized, which means that they contain more parameters than there are training data samples(Arora et al., 2018).

In this case, distributed methods may experience *communication bottleneck*: the communication cost for the workers to transfer sensitive information in joint optimization can exceed by multiple orders of magnitude the cost of local computation(Dutta et al., 2020). One of the practical methods to transfer information more efficiently is to apply a local compression operator (Seide et al., 2014; Suresh et al., 2017; Konečný & Richtárik, 2018; Zhang et al., 2017) to the model's parameters (gradients or tensors) communicated across different clients.

**Definition 1.** A (possibly randomized) mapping $\mathcal{C} : \mathbb{R}^d \to \mathbb{R}^d$ is called a compression operator if transmission of compressed tensor $\mathcal{C}(x)$ incurs less communication cost than the transfer of initial tensor $x$.

Although compression decreases the number of bits transferred during each communication cycle, it also introduces extra compression errors. As a result, the number of rounds necessary to obtain a solution with the appropriate accuracy typically increases. However, as the trade-off frequently appears to favor compression over no compression, compression has been proven to be effective in practice.

Distributed Compressed Gradient Descent (DCGD) (Khirirat et al., 2018) provides the simplest and universal mechanism for distributed communication-efficient training with compression. With the given learning rate $\gamma$, DCGD implements the following update rule

$$x^{t+1} = x^t - \gamma\frac{1}{n}\sum_{i=1}^{n} g_i^t, \quad g_i^t = \mathcal{M}_i^t(\nabla f_i(x^t)). \tag{2}$$

Here, $g_i^t$ represents an estimated gradient, the result of the mapping of original dense and high-dimensional gradient $\nabla f_i(x^t) \in \mathbb{R}^d$ into a vector of same size that can be transferred efficiently with far fewer bits via $\mathcal{M}_i^t$ compression mechanism.

In some cases (Tang et al., 2020; Philippenko & Dieuleveut, 2020; Fatkhullin et al., 2021), it is desirable to add compression on the server side to have efficient communication between the server and clients in both directions. One could easily extend the general framework of DCGD to the case of bidirectional compression. If we define the general master compression mechanism as $\mathcal{M}^{M,t}$ and the worker compression mechanism as $\mathcal{M}_i^{W,t}$, we could formally write the general bidirectional DCGD algorithm as Algorithm 1.

## 2 Motivation and Background

Our primary motivation in this work is to design a compression mechanism that can vary the level of gradient compression depending on a local optimality condition. This section of the paper introduces several key concepts essential for the proposed adaptive scheme. We describe constant contractive compressors, discuss what adaptive compressors already exist in the literature, and rehash a lazy aggregation mechanism – the precursor for our adaptive compression.

### 2.1 Constant contractive compressors

Most methods employing gradient compression mechanisms use a compressor with a constant compression level. In this approach (Beznosikov et al., 2020; Khirirat et al., 2018), one sets

$$\mathcal{M}_i^t(x) \equiv \mathcal{C}(x), \tag{3}$$

where $\mathcal{C} : \mathbb{R}^d \to \mathbb{R}^d$ is a compression operator. There are two large classes of compression operators (or compressors) widely presented in the literature: i) *unbiased* compression operators and ii) *biased* or *contractive* compression operators. Our focus in this paper is on biased operators, the definition of which we provide below.

**Definition 2** (Biased or contractive compression operator). A mapping $\mathcal{C} : \mathbb{R}^d \to \mathbb{R}^d$ is called *biased* or *contractive* compression operator if there exists $0 < \alpha \leq 1$ such that

$$\mathbb{E}\left[\|\mathcal{C}(x) - x\|^2\right] \leq (1 - \alpha)\|x\|^2, \qquad \forall x \in \mathbb{R}^d. \tag{4}$$

Top-$K$ (Alistarh et al., 2018) and adaptive random (Beznosikov et al., 2020) sparsification compressors are typical examples of contractive compressors. Due to the biased nature of these compressors, until recently, there was a gap between the experimental and theoretical development of gradient descent methods based on contractive compressors. Thus, during the last years, algorithmic approaches have provided several methods of high practical importance, most notable of which is the so-called error feedback mechanism (Seide et al., 2014), fixing a divergence issue that appeared in practice and theory (Beznosikov et al., 2020). In contrast, in the theoretical development, until very recently, analytical studies offered weak sublinear (Stich et al., 2018; Karimireddy et al., 2019; Horváth & Richtárik, 2021) convergence rates under, in some cases, strong unrealistic assumptions. Recently, Richtárik et al. (2021) fixed this by providing a novel algorithmic and theoretical development that recovers GD $\mathcal{O}(1/T)$ rates, with the analysis using standard assumptions only. Fatkhullin et al. (2021) subsequently extended the framework by including several algorithmic and theoretical extensions, such as bidirectional compression and client stochasticity. Despite these advances, there are still many challenges in the theoretical understanding of these methods. One such challenge is a lack of a thorough theoretical study of error feedback methods in a convex setting.

### 2.2 Existing adaptive compressors

Using a static compression level of the compressor for all clients could limitate the optimization framework's capability. Conversely, adjusting the compression level for every client could help reduce overall training time, for example, in hardware heterogeneous cases (Horváth et al., 2021; Abdelmoniem & Canini, 2021).Ideally, the optimizer should be able to define the particular compression level for each client separately based on the local client's data.

Despite the significant practical interest in developing such methods, there is currently little research and understanding of adaptive mechanisms of this type. Only a few papers (Qu et al., 2021; Hönig et al., 2021; Mishchenko et al., 2022; Alimohammadi et al., 2022) provide convergence guarantees with explicit rates, and most of them are designed for the specific type of compressors only, mostly quantizers (Guo et al., 2020; Wen et al., 2017). So, in (Jhunjhunwala et al., 2021), the authors design a mechanism for adaptive change of quantization level $s^k \sim \sqrt{\frac{f(x^0)}{f(x^k)}}$ of a random dithering operator (Alistarh et al., 2017). DAdaQuant (Hönig et al., 2021) and FedDQ (Qu et al., 2021) suggest ascending and descending quantizations throughout the training. AQUILA (Zhao et al., 2022) and AGQ (Mao et al., 2021) build an adaptive quantization on top of the Lazily Aggregated Quantized (LAQ) gradient algorithm (Sun et al., 2019). Faghri et al. (2020) proposes ALQ, an adaptive method for quantizing gradients that minimizes the excess variance of quantization given an estimate of the distribution of the gradients. It adapts individual quantization levels iteratively to

**Table 1** Summary of adaptive compressed methods. `AdaCGD`, proposed in this work, is superior to its counterparts in all three settings.

| Paper | Any $\mathcal{C}$? | Theory? | Strongly convex rate[5] | Convex rate | General nonconvex rate |
|---|---|---|---|---|---|
| Jhunjhunwala et al. (2021) | ✗ | ✗ | ✗ | ✗ | ✗ |
| Abdelmoniem & Canini (2021) | ✗ | ✗ | ✗ | ✗ | ✗ |
| Hönig et al. (2021) | ✗ | ✓[1] | $\frac{\max\{\kappa,\frac{\kappa^2}{n},\frac{n}{\mu^2}\}}{T^2}$ | ✗ | $\mathcal{O}(\frac{L\Delta_f}{\sqrt{T}}+\frac{C}{T})$ |
| Qu et al. (2021) | ✗ | ✓ | ✗ | ✗ | $\mathcal{O}(\frac{L\Delta_f}{\sqrt{T}})^{[2]}$ |
| Zhao et al. (2022) | ✗ | ✓ | linear [3] | ✗ | ✗ |
| Mao et al. (2021) | ✗ | ✓ | linear [3] | ✗ | ✗ |
| Khirirat et al. (2021) | ✓ | ✗ | ✗ | ✗ | ✗ |
| Alimohammadi et al. (2022) | ✗ | ✓ | ✗ | ✗ | $\mathcal{O}\left(\frac{\Delta_f}{\sqrt{Tn}}+\frac{L^2n}{T}+\frac{L^3n^{\frac{3}{2}}}{T^{\frac{3}{2}}}\right)^{[6]}$ |
| Zhong et al. (2021) | ✓ | ✓ | ✗ | ✗ | $\mathcal{O}(\frac{C}{\sqrt{T}})$ |
| Faghri et al. (2020) | ✗ | ✓ | ✗ | ✗ | $\mathcal{O}(\frac{C}{\sqrt{T}})$ |
| Mishchenko et al. (2022) | ✗ | ✓[4] | ✗ | $\mathcal{O}(\frac{L\Delta_x}{T}+\frac{\sigma_*^2+\varepsilon}{Ln})$ | $\mathcal{O}(\frac{L\Delta_f}{T}+\frac{\varepsilon}{Ln})$ |
| THIS WORK | ✓ | ✓ | $\left(1-\min\left\{\frac{\mu}{M_2},A_{\min}\right\}\right)^T$ | $\mathcal{O}\left(\frac{M_1}{T}\right)$ | $\mathcal{O}\left(\frac{2\Delta_f M_2+C/A_{\min}}{T}\right)$ |

[1] Rates, as suggested in the original paper, are derived from (Reisizadeh et al., 2020) and calculated for non-local full gradient setup ($\sigma^2 = 0$, $\tau = 1$).
[2] We show the rate for non-local full gradient setup, *i.e.* $\sigma^2 = 0$ and $\tau = 1$.
[3] Their work does not present any *explicit* rate.
[4] $\varepsilon > 0$ is a parameter of IntSGD algorithm.
[5] This column incorporates rates for the Polyak-Łojasiewicz nonconvex setting for its similarity with the strongly convex setting.
Notation: $\kappa = \frac{L}{\mu}$, $\{C_i\}_{i=1}^3$ are scalar constants, $\Delta_x = \|x^0 - x^*\|^2$, $\Delta_f = f(x_0) - f^*$, $M_1 = \max\{L_- + L_+\sqrt{\frac{2B_{\max}}{A_{\min}}}, \frac{1}{A_{\min}}\}$, $M_2 = L_- + L_+\sqrt{\frac{B_{\max}}{A_{\min}}}$ (see Lemma 2 for details).
[6] These rates were obtained for nonstandard compressor types $\|C(x) - x\|^2 \leq B$, where B is a constant value.

minimize the expected normalized variance. L-GRECO (Alimohammadi et al., 2022) dynamically adapts the degree of compression across the model's different layers during training using dynamic programming. IntSGD (Mishchenko et al., 2022) adaptively sets the scaling parameter $\alpha^k$ of a vector plugged in a randomized integer rounding operator. CLAN (Zhong et al., 2021) deals with constant contractive compressors and combines adaptive stepsize rule similar to ADAM(Kingma & Ba, 2017) with error-feedback applied on both server and client sides. The work of (Agarwal et al., 2021) presents Accordion, an adaptive compression algorithm that avoids over-compressing gradients in critical learning regimes, which can have a negative impact on model performance. The critical regime is detected by the rate of change in gradient norms. CAT S+Q (Khirirat et al., 2021) proposes an adaptive way to choose $k$: the top-$k$ elements of the gradient at iteration $i$, only *signs* of which clients send to the server along with the gradient norm. Table 1 provides a detailed comparison of these works.

### 2.3 Adaptive compression via selective (*lazy*) aggregation

We revisit the `LAG` mechanism, proposed by Chen et al. (2018), and consider it an alternative way to embed adaptivity into the framework by introducing communication skipping. According to the lazy aggregation communication scheme, each worker $i$ only shares its local gradient if it is significantly different from the last gradient shared previously. Otherwise, the worker decides to skip the communication round. One can consider `LAG` as an *adaptive* mechanism that chooses between two extremes: sending a full gradient or skipping the communication round.

Richtárik et al. (2022) recently extended this idea by introducing the `CLAG` algorithm that dispatches a *compressed* update when an old gradient estimate differs too much from the gradient at a new iteration or skips the communication at all. At each iteration of the algorithm, worker $i$ updates its gradient estimate $g_i^{t+1}$ by the following rule:

$$g_i^{t+1} = \begin{cases} g_i^t + \mathcal{C}(\nabla f_i(x^{t+1}) - g_i^t) & \text{if } \|\nabla f_i(x^{t+1}) - g_i^t\|^2 > \zeta\|\nabla f_i(x^{t+1}) - \nabla f_i(x^t)\|^2, \\ g_i^t & \text{otherwise,} \end{cases} \tag{5}$$

where $\zeta > 0$ is a trigger parameter, $g_i^t$ is a gradient estimate at previous iteration, and $\mathcal{C}$ is a biased compression operator. When the condition in the first line, called a *trigger condition*, fires, worker $i$ sends an update $\mathcal{C}(\nabla f_i(x^{t+1}) - g_i^t)$ to the server that smartly aggregates updates from workers and gets new full gradient estimate $g^{t+1} = (1/n)\sum_{i=1}^n g_i^{t+1}$. When the trigger condition does not fire, worker $i$ skips communication, which means $g_i^{t+1} = g_i^t$. Trigger parameter

**Table 2** Comparison of convergence guarantees results of methods employing lazy aggregation.

| Method | Adaptive compression? | Bidirectional case | Str cvx case | Cvx case | PŁ noncvx case | General noncvx case |
|---|---|---|---|---|---|---|
| LAQ (Sun et al., 2019) | ✗ | ✗ | ✓ | ✗ | ✗ | ✗ |
| LENA (Ghadikolaei et al., 2021) | ✗ | ✗ | ✓ | ✓ | ✓ | ✓ |
| LAG (Richtárik et al., 2022) | ✗ | ✗ | ✓ | ✗ | ✓ | ✓ |
| CLAG (Richtárik et al., 2022) | ✗ | ✗ | ✓ | ✗ | ✓ | ✓ |
| AdaCGD (NEW, 2022) | ✓ | ✓ | ✓ | ✓ | ✓ | ✓ |

$\zeta$ controls how frequently the trigger condition will fire, *i.e.*, how often clients skip communication. The update rule in Equation (5) also could be seen as an adaptive compression switching between two extremes: compressing at a pre-defined compression level or compressing at the maximum possible level, in other words, not sending anything at all.

Although both LAG and CLAG perform well in practice, their fixed and limited compression levels could restrict their performance and make these methods sub-optimal. It is of particular practical interest to create a more general method with evolving fine-tuned compression levels.

## 3 Summary of Contributions

We highlight our main contributions as follows:

● **New class of adaptive compressors.** In (Richtárik et al., 2022), the authors propose the different classes of compressors unified through a single theory. Despite the large variability of the compression mechanisms, including the algorithms with *lazy* aggregation rule, the compression level in all of the considered algorithms is pre-defined and constant during the training. In this work, we take a step further and formulate an extended class of adaptive 3PC compressors (Ada3PC) with tunable compression levels defined by some general trigger rules. As original 3PC compressors, this class of compressors is very general and includes several specific compressors such as AdaCGD, which includes EF21 and CLAG as special cases. We build Ada3PC compressors upon a large class of biased compressors, such as Top-$K$ and adaptive random sparsification.

● **Convergence guarantees with strong rates unified by a single 3PC theory.** We provide a strong convergence bound for strongly convex, convex, and nonconvex settings. Compared with the adaptive methods outside the 3PC context, we provide a more elaborate theory with better convergence rates. For AdaCGD, we recover $\mathcal{O}(1/T)$ rate of GD up to a certain constant in general nonconvex case. It is superior in comparison with $\mathcal{O}(1/\sqrt{T})$ rate (Hönig et al., 2021; Qu et al., 2021) for SOTA in adaptive compression outside 3PC context. The convergence theory in a convex set is of particular interest due to its novelty even in the case of 3PC for some key cases of AdaCGD, such as EF21 and CLAG. In other words, it is a new SOTA result for the error-feedback method in the convex setting.

● **Extension of 3PC theory with bidirectional compression.** We extend 3PC methods with bidirectional compression, *i.e.*, we allow the server to compress as well.

Table 2 compares AdaCGD with other described in the literature lazy algorithms.

## 4 Ada3PC: A Compression-Adaptive 3PC Method

### 4.1 3PC compressor

Richtárik et al. (2022) introduces the general concept of three point compressors. Here we provide its formal definition for consistency:

**Definition 3.** We say that a (possibly randomized) map $\mathcal{C}_{h,y}(x) : \underbrace{\mathbb{R}^d}_{h\in} \times \underbrace{\mathbb{R}^d}_{y\in} \times \underbrace{\mathbb{R}^d}_{x\in} \to \mathbb{R}^d$ is a three point compressor

(3PC) if there exist constants $0 < A \le 1$ and $B \ge 0$ such that the following relation holds for all $x, y, h \in \mathbb{R}^d$

$$\mathbb{E}\left[\|\mathcal{C}_{h,y}(x) - x\|^2\right] \le (1-A)\|h-y\|^2 + B\|x-y\|^2. \tag{6}$$

Authors show that `EF21` compression mechanism satisfies Definition 3. Let $\mathcal{C} : \mathbb{R}^d \to \mathbb{R}^d$ be a contractive compressor with contraction parameter $\alpha$, and define

$$\mathcal{C}_{h,y}^{\mathrm{EF}}(x) := h + \mathcal{C}(x - h). \tag{7}$$

If we use this mapping to define a compression mechanism $\mathcal{M}_i^t$ via (2) within `DCGD`, we obtain the `EF21` method.

Another variant of `3PC` compressors introduced in (Richtárik et al., 2022) is `CLAG` compressor. Let $\mathcal{C} : \mathbb{R}^d \to \mathbb{R}^d$ be a contractive compressor with contraction parameter $\alpha$. Choosing a trigger $\zeta > 0$, authors define the `CLAG` rule

$$\mathcal{C}_{h,y}^{\mathrm{CL}}(x) := \begin{cases} h + \mathcal{C}(x - h), & \text{if } \|x - h\|^2 > \zeta \|x - y\|^2, \\ h, & \text{otherwise,} \end{cases} \tag{8}$$

If we employ this mapping into `DCGD` method (2) as communication mechanism $\mathcal{M}_i^t$, we obtain `CLAG`. It includes `LAG` compressor $\mathcal{C}^{\mathrm{L}}$ as a special case when compressor $\mathcal{C}$ is identity.

## 4.2 Adaptive 3PC compressor

We are ready to introduce the Adaptive Three Point (`Ada3PC`) Compressor.

**Definition 4** (`Ada3PC` compressor). Let $\mathcal{C}^1, \mathcal{C}^2, \ldots, \mathcal{C}^m$ be 3PC compressors: $\mathcal{C}^i : \mathbb{R}^{3d} \to \mathbb{R}^d$ for all $i$. Let $P_1, P_2, \ldots, P_{m-1}$ be conditions depending on $h, y, x$, i.e. $P_j : \underbrace{\mathbb{R}^d}_{h\in} \times \underbrace{\mathbb{R}^d}_{y\in} \times \underbrace{\mathbb{R}^d}_{x\in} \to \{0, 1\}$ for all $j$. Then, the adaptive 3PC compressor, associated with the above 3PC compressors and conditions, is defined as follows:

$$\mathcal{C}_{h,y}^{\mathrm{ad}}(x) = \begin{cases} \mathcal{C}_{h,y}^1(x) & \text{if } P_1(h, y, x), \\ \mathcal{C}_{h,y}^2(x) & \text{else if } P_2(h, y, x), \\ \ldots, \\ \mathcal{C}_{h,y}^{m-1}(x) & \text{else if } P_{m-1}(h, y, x), \\ \mathcal{C}_{h,y}^m(x) & \text{otherwise.} \end{cases} \tag{9}$$

`Ada3PC` compressor first finds the smallest index $j$ for which $P_j(h, y, x) = 1$ (if such index does not exist, we set $j = m$). Then, `Ada3PC` applies $\mathcal{C}^j$ compressor on vector $x$.

## 4.3 Adaptive Compressed Gradient Descent

There are many ways to define meaningful and practical compressors in the context of the adaptive `3PC` framework. Here we provide one particular, perhaps the simplest scheme, which we define as `AdaCGD`. In this scheme, we have a set of $m$ `EF21` compressors $\{\mathcal{C}_{h,y}^{\mathrm{EF},j}(x)\}_{j\in 1\ldots m}$ sorted in order from the highest compression level to the lowest, i.e. $\alpha_1 \leq \alpha_2 \ldots \leq \alpha_m$, where $\alpha_j$ is a corresponding contractive parameter of the $j$-th compressor. For example, if we use Top-$K$ in $\mathcal{C}_{h,y}^{\mathrm{EF}}$ compressors, the first and last indices correspond to the compressors with the smallest and the largest $K$, respectively. We choose the first compressor, i.e. with the strongest compression, which satisfies a trigger rule. We design the $j$-th trigger rule following an intuition of *lazy aggregation* rule:

$$P_j := \|x - \mathcal{C}_{h,y}^{\mathrm{EF},j}(x)\|^2 \leq \zeta \|x - y\|^2. \tag{10}$$

As in the original `LAG` rule, the left side of (10) presents the difference between the actual gradient and its estimate, while the right side compares gradient differences on neighboring iterations. The key difference of (10) trigger from `LAG` and `CLAG` rule (5) is that the left side of this trigger condition depends explicitly on the level of compression. This feature is essential as it enables the desired rule-based compressor selection. Altogether, we can define `AdaCGD` compressor formally.

**Definition 5** (AdaCGD compressor). *Given the set of $m$ EF21 compressors $\{\mathcal{C}_{h,y}^{\mathrm{EF},j}(x)\}_{j\in 1\ldots m}$, sorted in descending order of compression level and $\zeta \geq 0$, adaptive AdaCGD compressor is defined with a switch condition as follows:*

$$
\mathcal{C}_{h,y}^{\mathrm{AC}}(x) = \begin{cases}
h & \text{if } \|x-h\|^2 \leq \zeta\|x-y\|^2, \\
\mathcal{C}_{h,y}^{\mathrm{EF},1}(x) & \text{else if } \|x-\mathcal{C}_{h,y}^{\mathrm{EF},1}(x)\|^2 \leq \zeta\|x-y\|^2, \\
\ldots, & \\
\mathcal{C}_{h,y}^{\mathrm{EF},m-1}(x) & \text{else if } \|x-\mathcal{C}_{h,y}^{\mathrm{EF},m-1}(x)\|^2 \leq \zeta\|x-y\|^2, \\
\mathcal{C}_{h,y}^{\mathrm{EF},m}(x) & \text{otherwise.}
\end{cases}
\tag{11}
$$

If $\mathcal{C}_{h,y}^{\mathrm{EF},m}$ uses Top-$d$ compression, *i.e.*, identity operator, AdaCGD is an adaptive compressor composed of the whole spectrum of compressors from full compression, *i.e.*, communication "skip", to zero compression, *i.e.*, sending full gradient. Since the standalone "skip" connection is not 3PC operator, it may not be evident that AdaCGD is a special case of Ada3PC. For this reason, here we provide the following lemma:

**Lemma 1.** *AdaCGD is a special case of Ada3PC compressor.*

It is easy to see that if $\zeta = 0$ AdaCGD reduces to EF21. Similarly, CLAG is a special case of AdaCGD when $m = 1$.

## 5 Theory

In this section, we present theoretical convergence guarantees for Algorithm 1 with Ada3PC compressors in two new cases presented in Table 2. The results for general and PŁ nonconvex cases can be found in the appendix.

### 5.1 Assumptions

We rely on standard assumptions to get convergence rates of Algorithm 1.

**Assumption 1.** *The functions $f_1, \ldots, f_n : \mathbb{R}^d \to \mathbb{R}$ are convex, i.e.*

$$
f_i(x) - f_i(y) - \langle \nabla f_i(y), x-y \rangle \geq 0, \ \forall x,y \in \mathbb{R}^d, \forall i.
\tag{12}
$$

**Assumption 2.** *The function $f : \mathbb{R}^d \to \mathbb{R}$ is $L_-$-smooth, i.e.*

$$
\|\nabla f(x) - \nabla f(y)\| \leq L_-\|x-y\|, \ \forall x,y \in \mathbb{R}^d.
\tag{13}
$$

**Assumption 3.** *There exists $L_+ > 0$ such that $\frac{1}{n}\sum_{i=1}^n \|\nabla f_i(x) - \nabla f_i(y)\|^2 \leq L_+^2\|x-y\|^2$ for all $x,y \in \mathbb{R}^d$. Let $L_+$ be the smallest such number.*

We borrow $\{L_-, L_+\}$ notation from (Szlendak et al., 2022). Assumption 3 avoids a stronger assumption on Lipschitz smoothness of individual functions $f_i$. Moreover, one can easily prove that $L_- \leq L_+$.

The next assumption is standard for analysis of practical methods (Ahn et al., 2020), Rajput et al. (2020). However, compared to previous works, we require a more general version.

**Assumption 4.** *We assume that there exists a constant $\Omega > 0$ such that $\mathbb{E}[\|x^t - x^*\|^2] \leq \Omega^2$, where $x^t$ is any iterate generated by Algorithm 1.*

**Assumption 5.** *The functions $f_1, \ldots, f_n$ are differentiable. Moreover, $f$ is bounded from below by an infimum $f^{\mathrm{inf}} \in \mathbb{R}$, i.e. $f(x) \geq f^{\mathrm{inf}}$ for all $x \in \mathbb{R}^d$.*

### 5.2 Adaptive 3PC is a 3PC compressor

While this may not be obvious at first glance, Adaptive 3PC compressors belong to the class of 3PC compressors. We formalize this statement in the following lemma.

**Lemma 2.** *Let $\mathcal{C}^{ad}$ be an adaptive 3PC compressor. Let $\{\mathcal{C}^i\}_{i=1}^m$ be 3PC compressors associated with $\mathcal{C}^{ad}$, i.e. for all $i$ there exists constants $0 < A_i \leq 1$ and $B_i \geq 0$, such that for all $h, y, x \in \mathbb{R}^d$*

$$
\mathbb{E}\|C_{h,y}^i(x) - x\|^2 \leq (1-A_i)\|h-y\|^2 + B_i\|x-y\|^2.
$$

*Then, $\mathcal{C}^{ad}$ is a 3PC compressor with constants $A_{\min} := \min\{A_1, \ldots, A_m\}$ and $B_{\max} := \max\{B_1, \ldots, B_m\}$.*

*Proof.* Let us fix $h, y, x \in \mathbb{R}^d$ and let $j$ be the index, such that $P_i(h, y, x) = 0$ for all $i < j$ and, if $j < m$, $P_j(h, y, x) = 1$. Then,

$$\mathbb{E}\|\mathcal{C}_{h,y}^{\mathrm{ad}}(x) - x\|^2 = \mathbb{E}\|\mathcal{C}_{h,y}^{j}(x) - x\|^2 \overset{(6)}{\leq} (1 - A_j)\|h - y\|^2 + B_j\|x - y\|^2$$
$$\leq (1 - A_{\min})\|h - y\|^2 + B_{\max}\|x - y\|^2.$$

$\square$

In the definition of Ada3PC compressor, we never specify what conditions $P_i$s are. They are completely arbitrary! The latter enables us to build infinitely many new compressors out of a few notable examples, presented in (Richtárik et al., 2022).

## 5.3 Convergence

In this subsection, we show how assumptions we make about minimized functions and compressors affect the convergence rate of Algorithm 1.

**Convergence for convex functions.** The first result assumes that $\mathcal{M}^{\mathrm{W}}$ in Algorithm 1 is a 3PC compressor, $\mathcal{M}^{\mathrm{M}}$ is an identity compressor: $\mathcal{M}^{\mathrm{M}}(x) = x \,\forall x \in \mathbb{R}^d$.

**Theorem 6.** *Let Assumptions 1, 2, 3 and 4 hold. In Algorithm 1, assume $\mathcal{M}^{\mathrm{W}}$ is a 3PC compressor, $\mathcal{M}^{\mathrm{M}}$ is an identity compressor, and the stepsize $\gamma$ satisfies $0 < \gamma \leq 1/M$, where $M = L_- + L_+\sqrt{\frac{2B}{A}}$. Then, for any $T \geq 1$ we have*

$$\mathbb{E}\left[f(x^T)\right] - f(x^\star) \leq \max\left\{\frac{1}{\gamma}, \frac{1}{A}\right\} \frac{2(\Omega^2 + \Phi^0)}{T},$$

*where $\Phi^t := f(x^t) - f(x^\star) + \frac{\gamma}{A}\frac{1}{n}\sum_{i=1}^{n}\|\nabla f_i(x^t) - g_i^t\|^2$ for any $t \geq 0$.*

The theorem combined with Lemma 2 implies the following fact.

**Corollary 1.** *Let the assumptions of Theorem 6 hold, assume $\mathcal{M}^{\mathrm{W}}$ is an Ada3PC compressor, $\mathcal{M}^{\mathrm{M}}$ is an identity compressor, and choose the stepsize $\gamma = \frac{1}{L_- + L_+\sqrt{\frac{2B_{\max}}{A_{\min}}}}$. Then, for any $T \geq 1$ we have*

$$\mathbb{E}\left[f(x^T)\right] - f(x^*) \leq \max\left\{L_- + L_+\sqrt{\frac{2B_{\max}}{A_{\min}}}, \frac{1}{A_{\min}}\right\} \frac{2(\Omega^2 + \Phi^0)}{T}.$$

*Thus, to achieve $\mathbb{E}\left[f(x^T)\right] - f(x^*) \leq \varepsilon$ for some $\varepsilon > 0$, the Ada3PC method requires*

$$T = \mathcal{O}\left(\max\left\{L_- + L_+\sqrt{\frac{2B_{\max}}{A_{\min}}}, \frac{1}{A_{\min}}\right\} \frac{2(\Omega^2 + \Phi_0^2)}{\varepsilon}\right)$$

*iterations.*

**Convergence for bidirectional method.** Here, we analyze the case when meaningful compressors are applied on both communication directions, *i.e.*, both $\mathcal{M}^{\mathrm{M}}$ and $\mathcal{M}^{\mathrm{W}}$ are 3PC compressors.

**Theorem 7.** *Let Assumptions 3 and 5 hold. Let $\mathcal{M}^{\mathrm{M}}$ and $\mathcal{M}^{\mathrm{W}}$ be 3PC compressors and the stepsize in Algorithm 1 be set as*

$$0 < \gamma \leq \frac{1}{L_- + L_+\sqrt{\frac{6B^M(B^W+1)}{A^M} + \frac{2B^W}{A^M}\left(1 + \frac{3B^M(2-A^W)}{A^M}\right)}}. \tag{14}$$

*Fix $T$ and let $\hat{x}^T$ be chosen uniformly from $\{x^0, x^1, \cdots, x^{T-1}\}$ uniformly at random. Then*

$$\mathbb{E}\left[\left\|\nabla f(\hat{x}^T)\right\|^2\right] \leq \frac{2\Psi^0}{\gamma T}. \tag{15}$$

*where $\Psi^t = f(x^t) - f^{\mathrm{inf}} + \frac{\gamma}{A^M}\|g^t - \tilde{g}^t\|^2 + \frac{\gamma}{A^W}\left(1 + \frac{3B^M(2-A^W)}{A^M}\right)\frac{1}{n}\sum_{i=1}^{n}\|\tilde{g}_i^t - \nabla f_i(x^t)\|^2$ for any $t \geq 0$.*

In the theorem, superscripts "M" and "W" denote master and worker compressor parameters, respectively. Theorem 7 implies the following corollary.

**Corollary 2.** *Let the assumptions of Theorem 7 hold, assume $\mathcal{M}^M$ and $\mathcal{M}^W$ are* Ada3PC *compressors and the stepsize*

$$\gamma = \frac{1}{L_- + L_+ \sqrt{\frac{6B_{max}^M(B_{max}^W+1)}{A_{min}^M} + \frac{2B_{max}^W}{A_{min}^M}\left(1 + \frac{3B_{max}^M(2-A_{min}^W)}{A_{min}^M}\right)}}.$$

*Fix $T$ and let $\hat{x}^T$ be chosen uniformly from $\{x^0, x^1, \cdots, x^{T-1}\}$ uniformly at random. Then, we have*

$$\mathbb{E}\left[\left\|\nabla f(\hat{x}^T)\right\|^2\right] \leq \frac{2\Psi^0\left(L_- + L_+ \sqrt{\frac{6B_{max}^M(B_{max}^W+1)}{A_{min}^M} + \frac{2B_{max}^W}{A_{min}^M}\left(1 + \frac{3B_{max}^M(2-A_{min}^W)}{A_{min}^M}\right)}\right)}{T}.$$

*Thus, to achieve $\mathbb{E}\left[\|\nabla f(\hat{x}^T)\|^2\right] \leq \varepsilon^2$ for some $\varepsilon > 0$, Algorithm 1 requires*

$$T = \mathcal{O}\left(\frac{2\Psi^0\left(L_- + L_+ \sqrt{\frac{6B_{max}^M(B_{max}^W+1)}{A_{min}^M} + \frac{2B_{max}^W}{A_{min}^M}\left(1 + \frac{3B_{max}^M(2-A_{min}^W)}{A_{min}^M}\right)}\right)}{T}\right)$$

*iterations.*

# 6 Experiments

In this work, we use the similar setup described in (Richtárik et al., 2022). Namely, we aim to solve the logistic regression problem with a nonconvex regularizer:

$$\min_{x \in \mathbb{R}^d}\left[f(x) := \frac{1}{N}\sum_{i=1}^N \log(1 + e^{-y_i a_i^\top x}) + \lambda \sum_{j=1}^d \frac{x_j^2}{1+x_j^2}\right],$$

where $a_i \in \mathbb{R}^d$, $y_i \in \{-1, 1\}$ are the training samples and labels with regularization hyperparameter $\lambda > 0$ are chosen at $\lambda = 0.1$ level. In training we use LIBSVM Chang & Lin (2011) datasets *phishing, a1a, a9a*. Each dataset has been split into $n = 20$ equal parts, each representing a different client.

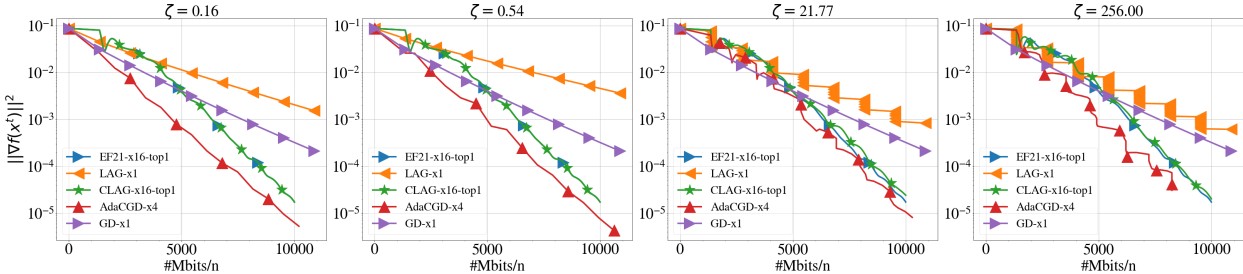

Figure 1: Comparison of LAG, CLAG, EF21 and GD with AdaCGD on phishing dataset. $1\times, 2\times, 4\times$ (and so on) indicates the multiplication factor we use for the optimal stepsizes predicted by theory. For example, GD-x1 means gradient decent (GD) with theoretical stepsize multiplied by 1.

Figure 1 compares the performance of our proposed algorithm, AdaCGD, with other popular 3PC methods. In order to make a fair comparison, we fine-tuned the stepsize of each considered algorithm with a set of multiples of the corresponding theoretical stepsize, ranging from $2^0$ to $2^8$. We implemented all simulations in Python 3.8, and ran them on a cluster of 48 nodes with Intel(R) Xeon(R) Gold 6230R CPUs. To evaluate the effectiveness of the contractive compressor used in each algorithm, we chose the Top-$k$ operator as our compressor of choice. For EF21 and CLAG,

we used the top-1 compressor, which has been shown to be the best in practice for these methods. For AdaCGD, we chose an array of compressors that varied from full compression (skip communication) to zero compression (sending the full gradient), with a step of 5. We used the communication cost of the algorithm as the stopping criterion for all experiments.

Our findings indicate that AdaCGD achieves comparable performance to CLAG, and in certain cases (notably the *phishing* dataset), it outperforms CLAG. Additionally, AdaCGD consistently outperforms LAG in all experimental settings. However, it is worth noting that on datasets such as *a1a* and *a9a*, where higher compression is more advantageous, CLAG exhibits slightly better convergence rates. These results highlight the valuable role of AdaCGD as a complement to existing 3PC methods, with the potential to enhance the convergence rate of distributed optimization algorithms. For further experimental details and results, please refer to the appendix.

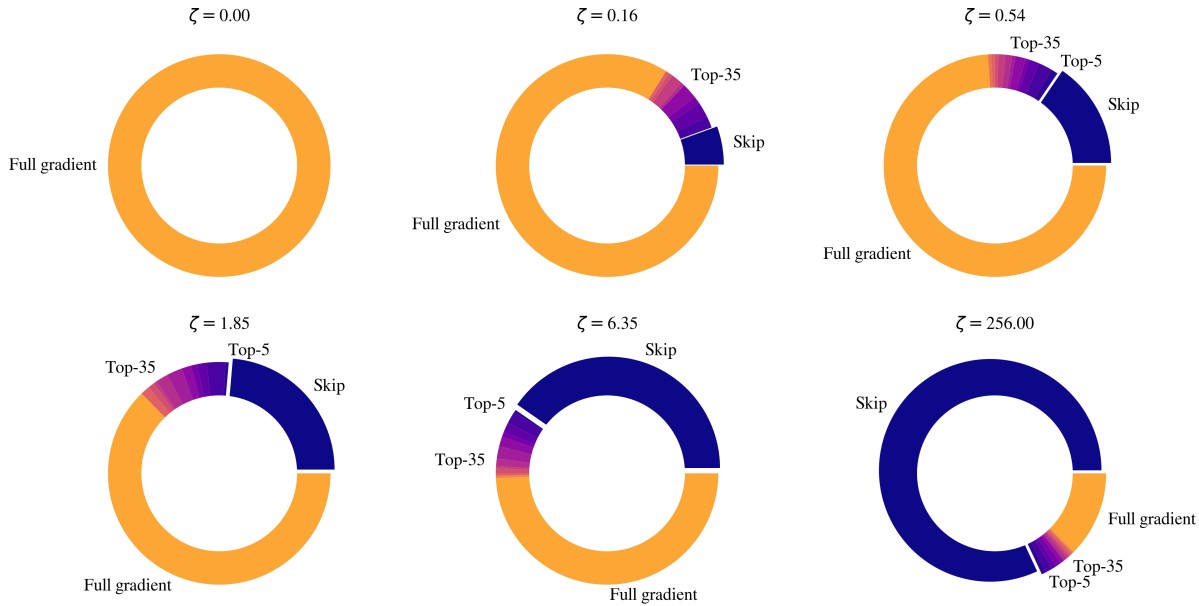

Figure 2: Distribution of compressors utilized during the training process on the phishing dataset for different $\zeta$ values, shown as multiple pie charts. The setting is the same as for Figure 1.

Figure 2 provides insights into how adaptivity in compression is utilized during the training process. The plot displays the distribution of compressors that were selected during training by all clients. The intuition behind the use of adaptivity is that when there is a smaller gap between neighboring time step gradients, it leads to smaller updates, allowing AdaCGD to apply larger compression without losing information. Conversely, when there is a larger gap between gradients, it means more informative updates that are more susceptible to higher compression, hence requiring AdaCGD to compress less. As expected, when $\zeta = 0$, AdaCGD is equivalent to EF21, which tends to choose the maximum compression level across different compressors. Conversely, with a larger $\zeta$ value, the AdaCGD behaves more like LAG/CLAG and has an increased number of skip connections. However, there is a range of $\zeta$ values (0.16, 0.54, 1.85, and 6.35) where the algorithm benefits from adaptivity and utilizes the full spectrum of compressors.

## 7   Discussion and Limitations

The main limitations of the work are assumptions we make upon functions $f_i$ of the problem 1. However, on the other hand, these assumptions govern the convergence rates we report: for example, we cannot show linear rates for convex functions due to the fundamental lower bound (Nesterov et al., 2018).

Another limitation comes from the analysis of the Bidirectional 3PC algorithm (Theorem 7). We show the analysis only for general nonconvex functions.

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

# APPENDIX

In Appendix A,we state the basic facts needed for detailed proofs of the propositions. In Appendix B, we provide the proofs missing in the main part of the paper. Appendix C contains experimental details and extra experiments. We briefly discuss the main limitations of the paper in Appendix D.

## A  Basic facts

We start the appendix with common math facts. Lemmas 3 and 4 present classic Cauchy-Schwartz inequality for vectors in metric space and random variables in probabilistic space, respectively. Lemma 5 shows a classic upper bound on quadratics. Lemma 6 provides a sufficient condition that ensures a quadratic inequality appearing in our convergence proofs holds.

**Lemma 3** (Cauchy-Schwartz inequality for arbitrary vectors). *Let $x, y \in \mathbb{R}^d$ be arbitrary vectors. Then, the following inequality holds*

$$|\langle x, y \rangle| \leq \|x\|\|y\|, \tag{16}$$

*where $\langle \cdot, \cdot \rangle$ and $\| \cdot \|$ denote the inner product and the induced norm, respectively.*

**Lemma 4** (Cauchy-Schwartz inequality for random variables; section 6.2.4 of (Pishro-Nik, 2014)). *For any two random variables $X$ and $Y$, we have*

$$|\mathbb{E}[XY]| \leq \sqrt{\mathbb{E}[X^2]\mathbb{E}[Y^2]}, \tag{17}$$

*where equality holds if and only if $X = \alpha Y$, for some constant $\alpha \in \mathbb{R}$.*

**Lemma 5.** *Let $a, b, c, d \in \mathbb{R}^d$ be arbitrary vectors. Then, the following inequalities hold*

$$\|a - b\|^2 \leq 2(\|a - c\|^2 + \|c - b\|^2), \tag{18}$$

$$\|a - b\|^2 \leq 3(\|a - c\|^2 + \|c - d\|^2 + \|d - b\|^2). \tag{19}$$

**Lemma 6** (Lemma 5 of (Richtárik et al., 2021)). *If $0 < \gamma \leq \frac{1}{\sqrt{a}+b}$, then $a\gamma^2 + b\gamma \leq 1$. Moreover, the bound is tight up to the factor of 2 since $\frac{1}{\sqrt{a}+b} \leq \min\{\frac{1}{\sqrt{a}}, \frac{1}{b}\} \leq \frac{2}{\sqrt{a}+b}$.*

# B  Proofs for Sections 4 and 5

## B.1  Lemma 1

At first glance, AdaCGD does not seem to be an Ada3PC compressor. However, we can construct an Ada3PC compressor, which is equivalent to AdaCGD.

**Lemma 1.** AdaCGD *is a special case of* Ada3PC *compressor.*

*Proof.* Let us consider the following Ada3PC compressor constructed from one LAG compressor and $m$ EF21 compressors.

$$
\mathcal{C}_{h,y}(x) = \begin{cases} \mathcal{C}_{h,y}^{\text{LAG}} = \begin{cases} h & \text{if } \|x - h\|^2 \le \zeta\|x - y\|^2, \\ x & \text{otherwise.} \end{cases} & \text{if } \|x - h\|^2 \le \zeta\|x - y\|^2, \\ \mathcal{C}_{h,y}^{\text{EF},1}(x) & \text{else if } \|x - \mathcal{C}_{h,y}^{\text{EF},1}(x)\|^2 \le \zeta\|x - y\|^2, \\ \dots, \\ \mathcal{C}_{h,y}^{\text{EF},m-1}(x) & \text{else if } \|x - \mathcal{C}_{h,y}^{\text{EF},m-1}(x)\|^2 \le \zeta\|x - y\|^2, \\ \mathcal{C}_{h,y}^{\text{EF},m}(x) & \text{otherwise.} \end{cases}
$$

If $\|x - h\|^2 \le \zeta\|x - y\|^2$, then $\mathcal{C}_{h,y}$ applies the LAG compressor to $x$. This LAG compressor in turn outputs $h$, as it does $\mathcal{C}_{h,y}^{\text{AC}}$ for the same condition. If the opposite is true, *i.e.*, $\|x - h\|^2 > \zeta\|x - y\|^2$, $\mathcal{C}_{h,y}$ checks the same conditions and chooses the same compressor as $\mathcal{C}_{h,y}^{\text{AC}}$. Thus, $\mathcal{C}_{h,y}^{\text{AC}}$ is equivalent to Ada3PC compressor $\mathcal{C}_{h,y}$. □

## B.2  Theorem 6

The proof of Theorem 6 requires several auxiliary results. Lemma 7 states the descent lemma typical for the analysis of biased compressors. Lemma 8 shows how individual 3PC compressors, applied at clients, affect the aggregated divergence of gradient estimates from gradients. Lemma 9 presents a technical upper bound on Lyapunov function $\Psi^t$.

**Lemma 7** (Lemma 2 of (Li et al., 2021))**.** *Suppose the function $f$ is $L_-$-smooth and $x^{t+1} = x^t - \gamma g^t$, where $g^t \in \mathbb{R}^d$ is any vector, and $\gamma > 0$ is any scalar. Then we have*

$$
f(x^{t+1}) - f(x^t) \le -\frac{\gamma}{2}\|\nabla f(x^t)\|^2 - \left(\frac{1}{2\gamma} - \frac{L_-}{2}\right)\|x^{t+1} - x^t\|^2 + \frac{\gamma}{2}\|g^t - \nabla f(x^t)\|^2. \tag{20}
$$

**Lemma 8** (Lemma B.3 of (Richtárik et al., 2022))**.** *Let Assumption 3 hold. Consider Algorithm 1 with 3PC compressor $\mathcal{M}^W$ and identity compressor $\mathcal{M}^M$. Then for all $t \ge 0$ the sequence*

$$
G^t = \frac{1}{n}\sum_{i=1}^n \|\nabla f_i(x^t) - g_i^t\|^2 \tag{21}
$$

*satisfies*

$$
\mathbb{E}\left[G^{t+1}\right] \le (1 - A)\mathbb{E}\left[G^t\right] + BL_+^2 \mathbb{E}\left[\|x^{t+1} - x^t\|^2\right], \tag{22}
$$

*where $A$ and $B$ are parameters of $\mathcal{M}^W$.*

**Lemma 9.** *Let Assumption 1 hold. Let Lyapunov function $\Psi^t := f(x^t) - f^* + \frac{\gamma}{A}G^t$. Then, for any $t \ge 0$, the following inequality holds*

$$
\mathbb{E}\Psi^t \le \sqrt{\left(\mathbb{E}\left[\|\nabla f(x^t)\|^2\right] + \frac{\gamma}{A}\mathbb{E}G^t\right)\left(\mathbb{E}\left[\|x^t - x^\star\|^2\right] + \frac{\gamma}{A}\mathbb{E}\left[G^t\right]\right)}, \tag{23}
$$

*where $x^*$ is any point belonging to* Argmin $f(x)$.

*Proof.* By definition of convexity we get

$$
\begin{aligned}
\mathbb{E}\Psi^t &= \mathbb{E}f(x^t) - f^* + \frac{\gamma}{A}\mathbb{E}G^t \\
&\overset{(12)}{\le} \mathbb{E}\langle\nabla f(x^t), x^t - x^\star\rangle + \frac{\gamma}{A}\mathbb{E}G^t \\
&= \mathbb{E}\left\langle\left[\nabla f(x^t), \sqrt{\frac{\gamma}{A}\mathbb{E}G^t}\right], \left[x^t - x^*, \sqrt{\frac{\gamma}{A}\mathbb{E}G^t}\right]\right\rangle
\end{aligned}
$$

.

By applying Cauchy-Schwartz inequality on vectors and random variables we finish the proof

$$\mathbb{E}\left\langle \left[\nabla f(x^t), \sqrt{\frac{\gamma}{A}\mathbb{E}G^t}\right], \left[x^t - x^*, \sqrt{\frac{\gamma}{A}\mathbb{E}G^t}\right]\right\rangle$$

$$\overset{(16)}{\leq} \mathbb{E}\left[\sqrt{\|\nabla f(x^t)\|^2 + \frac{\gamma}{A}\mathbb{E}G^t}\sqrt{\|x^t - x^\star\|^2 + \frac{\gamma}{A}\mathbb{E}G^t}\right]$$

$$\overset{(17)}{\leq} \sqrt{\left(\mathbb{E}\left[\|\nabla f(x^t)\|^2\right] + \frac{\gamma}{A}\mathbb{E}G^t\right)\left(\mathbb{E}\left[\|x^t - x^\star\|^2\right] + \frac{\gamma}{A}\mathbb{E}\left[G^t\right]\right)}.$$

$\square$

Now we are ready to prove the main theorem.

**Theorem 6.** *Let Assumptions 1, 2, 3 and 4 hold. Assume the stepsize $\gamma$ of algorithm satisfies $0 < \gamma \leq 1/M$, where $M = L_- + L_+\sqrt{\frac{2B}{A}}$. Then, for any $T \geq 0$ we have*

$$\mathbb{E}\left[f(x^T)\right] - f(x^\star) \leq \max\left\{\frac{1}{\gamma}, \frac{1}{A}\right\}\frac{2(\Omega^2 + \Psi^0)}{T}.$$

*Proof.* Combining Lemma 7, Jensen's inequality , we get

$$f(x^{t+1}) - f(x^t) \leq -\frac{\gamma}{2}\|\nabla f(x^t)\|^2 - \left(\frac{1}{2\gamma} - \frac{L_-}{2}\right)\|x^{t+1} - x^t\|^2 + \frac{\gamma}{2}\left\|\frac{1}{n}\sum_{i=1}^{n}g_i^t - \frac{1}{n}\sum_{i=1}^{n}\nabla f_i(x^t)\right\|^2$$

$$\leq -\frac{\gamma}{2}\|\nabla f(x^t)\|^2 - \left(\frac{1}{2\gamma} - \frac{L_-}{2}\right)\|x^{t+1} - x^t\|^2 + \frac{\gamma}{2}\frac{1}{n}\sum_{i=1}^{n}\|g_i^t - \nabla f_i(x^t)\|^2 \qquad (24)$$

$$= -\frac{\gamma}{2}\|\nabla f(x^t)\|^2 - \left(\frac{1}{2\gamma} - \frac{L_-}{2}\right)\|x^{t+1} - x^t\|^2 + \frac{\gamma}{2}G^t.$$

Now applying Equation (24) and Lemma 8 on the difference of Lyapunov functions, we obtain

$$\mathbb{E}\left[\Psi^{t+1}\right] - \mathbb{E}\left[\Psi^t\right] = \mathbb{E}\left[f(x^{t+1}) - f(x^t)\right] + \frac{\gamma}{A}\mathbb{E}\left[G^{t+1}\right] - \frac{\gamma}{A}\mathbb{E}\left[G^t\right]$$

$$\overset{(24)}{\leq} -\frac{\gamma}{2}\mathbb{E}\left[\|\nabla f(x^t)\|^2\right] - \left(\frac{1}{2\gamma} - \frac{L_-}{2}\right)\mathbb{E}\left[\|x^{t+1} - x^t\|^2\right] + \frac{\gamma}{2}\mathbb{E}\left[G^t\right]$$

$$+ \frac{\gamma}{A}\mathbb{E}\left[G^{t+1}\right] - \frac{\gamma}{A}\mathbb{E}\left[G^t\right]$$

$$\overset{(22)}{\leq} -\frac{\gamma}{2}\mathbb{E}\left[\|\nabla f(x^t)\|^2\right] - \left(\frac{1}{2\gamma} - \frac{L_-}{2}\right)\mathbb{E}\left[\|x^{t+1} - x^t\|^2\right]$$

$$+ \frac{\gamma}{A}\left[(1 - A)\mathbb{E}[G^t] + BL_+^2\mathbb{E}\|x^{t+1} - x^t\|^2 - \mathbb{E}[G^t]\right].$$

Rearranging the term, we get

$$\mathbb{E}\left[\Psi^{t+1}\right] - \mathbb{E}\left[\Psi^t\right] \leq -\frac{\gamma}{2}\left[\|\nabla f(x^t)\|^2\right] - \left(\frac{1}{2\gamma} - \frac{L_-}{2} - \frac{\gamma BL_+^2}{A}\right)\mathbb{E}\left[\|x^{t+1} - x^t\|^2\right] - \frac{A}{2}\frac{\gamma}{A}\mathbb{E}\left[G^t\right].$$

We further note that

$$\frac{1}{2\gamma} - \frac{L_-}{2} - \frac{\gamma BL_+^2}{A} \geq 0 \Leftrightarrow L_+^2\frac{2B}{A}\gamma^2 + L_-\gamma \leq 1 \overset{Lemma\ 6}{\Longleftarrow} \gamma \leq \frac{1}{L_- + L_+\sqrt{\frac{2B}{A}}}.$$

Appropriately chosen stepsize gives

$$\mathbb{E}\left[\Psi^{t+1}\right] - \mathbb{E}\left[\Psi^t\right] \leq -\min\left\{\frac{\gamma}{2}, \frac{A}{2}\right\}\left(\mathbb{E}\left[\|\nabla f(x^t)\|^2\right] + \frac{\gamma}{A}\mathbb{E}\left[G^t\right]\right).$$

Rearranging the terms, we have

$$\mathbb{E}\left[\|\nabla f(x^t)\|^2\right] + \frac{\gamma}{A}\mathbb{E}\left[G^t\right] \leq \frac{\mathbb{E}\left[\Psi^t\right] - \mathbb{E}\left[\Psi^{t+1}\right]}{\min\left\{\frac{\gamma}{2}, \frac{A}{2}\right\}}. \tag{25}$$

from what we deduce that $\mathbb{E}\left[\Psi^{t+1}\right] \leq \mathbb{E}\left[\Psi^t\right]$.

Using Lemma 9 and (25), we have

$$
\begin{aligned}
\mathbb{E}\Psi^{t+1}\mathbb{E}\Psi^t &\leq \left(\mathbb{E}\Psi^t\right)^2 \leq \left(\mathbb{E}\left[\|\nabla f(x^t)\|^2\right] + \frac{\gamma}{A}\mathbb{E}G^t\right)\left(\mathbb{E}\left[\|x^t - x^\star\|^2\right] + \frac{\gamma}{A}\mathbb{E}\left[G^t\right]\right) \\
&\leq \frac{\mathbb{E}\left[\|x^t - x^\star\|^2\right] + \frac{\gamma}{A}\mathbb{E}\left[G^t\right]}{\min\left\{\frac{\gamma}{2}, \frac{A}{2}\right\}}\left(\mathbb{E}\left[\Psi^t\right] - \mathbb{E}\left[\Psi^{t+1}\right]\right).
\end{aligned}
$$

Using that $\frac{\gamma}{A}\mathbb{E}\left[G^t\right] \leq \mathbb{E}\Psi^t \leq \Psi_0$ and $\mathbb{E}\left[\|x^t - x^\star\|^2\right] \leq \Omega^2$, we obtain

$$\mathbb{E}\Psi^{t+1}\mathbb{E}\Psi^t \leq \frac{\Omega^2 + \Psi^0}{\min\left\{\frac{\gamma}{2}, \frac{A}{2}\right\}}\left(\mathbb{E}\left[\Psi^t\right] - \mathbb{E}\left[\Psi^{t+1}\right]\right).$$

Rearranging again, we get

$$\frac{\min\left\{\frac{\gamma}{2}, \frac{A}{2}\right\}}{\Omega^2 + \Psi^0} \leq \left(\frac{1}{\mathbb{E}\left[\Psi^{t+1}\right]} - \frac{1}{\mathbb{E}\left[\Psi^t\right]}\right).$$

Summing up from $t = 0$ to $t = T - 1$, we finish the proof

$$\mathbb{E}\left[f(x^T)\right] - f(x^\star) \leq \mathbb{E}\left[\Psi^T\right] \leq \max\left\{\frac{2}{\gamma}, \frac{2}{A}\right\}\frac{\Omega^2 + \Psi^0}{T}. \tag{26}$$

$\square$

## B.3 Theorem 7

---
**Algorithm 2** 3PC-BD (Bidirectional 3PC algorithm)

---

1: **Input:** starting point $x^0 \in \mathbb{R}^d$; $g^0, \tilde{g}_i^0 \in \mathbb{R}^d$ for $i = 1, \cdots, n$ (known by nodes), $\tilde{g}^0 = \frac{1}{n}\sum_{i=1}^{n} \tilde{g}_i^0$ (known by master);
learning rate $\gamma > 0$.
2: **for** $t = 0,1,2,\cdots,T-1$ **do**
3:     Broadcast $g^t$ to all workers
4:     **for for all devices** $i = 1, \ldots, n$ **in parallel do**
5:         $x^{t+1} = x^t - \gamma g^t$
6:         $\tilde{g}_i^{t+1} = \mathcal{C}_{\tilde{g}_i^t, \nabla f_i(x^t)}^w(\nabla f_i(x^{t+1}))$
7:         Communicate $\tilde{g}_i^{t+1}$ to the server
8:     **end for**
9:     $\tilde{g}^{t+1} = \frac{1}{n}\sum_{i=1}^{n} \tilde{g}_i^{t+1}$
10:     $g^{t+1} = \mathcal{C}_{g^t, \tilde{g}^t}^M(\tilde{g}^{t+1})$
11: **end for**

---

For Theorem 7, we assume that both compressors $\mathcal{M}^W$ and $\mathcal{M}^M$ in Algorithm 1 are 3PC compressors. The main steps of the algorithm are:

$$x^{t+1} = x^t - \gamma g^t$$
$$\tilde{g}_i^{t+1} = \mathcal{C}_{\tilde{g}_i^t, \nabla f_i(x^t)}^w(\nabla f_i(x^{t+1}))$$
$$\tilde{g}^{t+1} = \frac{1}{n}\sum_{i=1}^n \tilde{g}_i^{t+1}$$
$$g^{t+1} = \mathcal{C}_{g^t, \tilde{g}^t}^M(\tilde{g}^{t+1})$$

Unlike in the previous subsection, we use additional notations: $P_i^t := \left\|\tilde{g}_i^t - \nabla f_i(x^t)\right\|^2$, $P^t := \frac{1}{n}\sum_{i=1}^n P_i^t$ and $R^t := \left\|x^{t+1} - x^t\right\|^2$.

Lemma 10 is an analogue of Lemma 8 (in bidirectional case we need slightly different arguments at some steps). Lemma 11 is another technical lemma that upper bounds $\mathbb{E}\left[\|g^t - \tilde{g}^t\|^2\right]$.

**Lemma 10.** *Let Assumption 3 hold, $\mathcal{C}^w$ be a 3PC compressor, and $\tilde{g}_i^{t+1}$ be an 3PC-BD estimator of $\nabla f_i(x^{t+1})$, i.e.*

$$\tilde{g}_i^{t+1} = \mathcal{C}_{\tilde{g}_i^t, \nabla f_i(x^t)}^w(\nabla f_i(x^{t+1})) \tag{27}$$

*for arbitrary $\tilde{g}_i^0$ for all $i \in [n], t \geq 0$. Then*

$$\mathbb{E}\left[P^{t+1}\right] \leq (1 - A^W)\mathbb{E}\left[P^t\right] + B^W L_+^2 \mathbb{E}\left[R^t\right]. \tag{28}$$

*Proof.* Define $W^t := \{\tilde{g}_1^t, \cdots, \tilde{g}_n^t, x^t, x^{t+1}\}$, then

$$
\begin{aligned}
\mathbb{E}\left[P_i^{t+1}\right] &= \mathbb{E}\left[\mathbb{E}\left[P_i^{t+1} \mid W^t\right]\right] \\
&= \mathbb{E}\left[\mathbb{E}\left[\left\|\tilde{g}_i^{t+1} - \nabla f_i(x^{t+1})\right\|^2 \mid W^t\right]\right] \\
&= \mathbb{E}\left[\mathbb{E}\left[\left\|\mathcal{C}_{\tilde{g}_i^t, \nabla f_i(x^t)}^w(\nabla f_i(x^{t+1})) - \nabla f_i(x^{t+1})\right\|^2 \mid W^t\right]\right] \\
&\overset{(6)}{\leq} (1 - A^W)\mathbb{E}\left[\left\|\tilde{g}_i^t - \nabla f_i(x^t)\right\|^2\right] + B^W \mathbb{E}\left[\left\|\nabla f_i(x^{t+1}) - \nabla f_i(x^t)\right\|^2\right]
\end{aligned} \tag{29}
$$

.

Averaging the above inequalities over $i \in [n]$, we obtain (28). Indeed,

$$
\begin{aligned}
\mathbb{E}\left[P^{t+1}\right] &= \mathbb{E}\left[\frac{1}{n}\sum_{i=1}^n P_i^{t+1}\right] = \frac{1}{n}\sum_{i=1}^n \mathbb{E}\left[P_i^{t+1}\right] \\
&\overset{(29)}{\leq} \frac{1}{n}\sum_{i=1}^n (1 - A^W)\mathbb{E}\left[\left\|\tilde{g}_i^t - \nabla f_i(x^t)\right\|^2\right] + \frac{1}{n}\sum_{i=1}^n B^W \mathbb{E}\left[\left\|\nabla f_i(x^{t+1}) - \nabla f_i(x^t)\right\|^2\right] \\
&= (1 - A^W)\mathbb{E}\left[P^t\right] + B^W \frac{1}{n}\sum_{i=1}^n \mathbb{E}\left[\left\|\nabla f_i(x^{t+1}) - \nabla f_i(x^t)\right\|^2\right] \\
&\overset{\text{Assumption 3}}{\leq} (1 - A^W)\mathbb{E}\left[P^t\right] + B^W L_+^2 \mathbb{E}\|x^{t+1} - x^t\|^2 \\
&= (1 - A^W)\mathbb{E}\left[P^t\right] + B^W L_+^2 \mathbb{E}\left[R^t\right].
\end{aligned}
$$

$\square$

**Lemma 11.** *Let Assumptions 3 and 5 hold, $\mathcal{C}^M, \mathcal{C}^w$ be 3PC compressors. Let $\tilde{g}_i^{t+1}$ be an 3PC-BD estimator of $\nabla f_i(x^{t+1})$, i.e.*

$$\tilde{g}_i^{t+1} = \mathcal{C}_{\tilde{g}_i^t, \nabla f_i(x^t)}^w(\nabla f_i(x^{t+1})) \tag{30}$$

*and let $g^{t+1}$ be an* 3PC-BD *estimator of $\tilde{g}^{t+1} = \frac{1}{n} \sum_{i=1}^{n} \tilde{g}_i^{t+1}$, i.e.*

$$g_i^{t+1} = \mathcal{C}_{g^t,\tilde{g}^t}^M(\tilde{g}^{t+1}) \tag{31}$$

*for arbitrary $g^0, \tilde{g}_i^0$ for all $i \in [n], t \geq 0$. Then*

$$\mathbb{E}\left[\left\|g^{t+1} - \tilde{g}^{t+1}\right\|^2\right] \leq (1 - A^M)\mathbb{E}\left[\left\|g^t - \tilde{g}^t\right\|^2\right] + 3B^M(2 - A^W)\mathbb{E}\left[P^t\right] + 3B^M(B^W + 1)L_+^2\mathbb{E}\left[R^t\right], \tag{32}$$

*where $g^t = \frac{1}{n} \sum_{i=1}^{n} g_i^t, \tilde{g}^t = \frac{1}{n} \sum_{i=1}^{n} \tilde{g}_i^t$.*

*Proof.* Similarly to the proof of Lemma 10, we define $W^t := \{g_1^t, \cdots, g_n^t, x^t, x^{t+1}\}$ and bound $\mathbb{E}\left[\left\|g^{t+1} - \tilde{g}^{t+1}\right\|^2\right]$:

$$
\begin{aligned}
\mathbb{E}\left[\left\|g^{t+1} - \tilde{g}^{t+1}\right\|^2\right] &= \mathbb{E}\left[\mathbb{E}\left[\left\|g^{t+1} - \tilde{g}^{t+1}\right\|^2 \mid W^t\right]\right] \\
&= \mathbb{E}\left[\mathbb{E}\left[\left\|\mathcal{C}_{g^t,\tilde{g}^t}^M(\tilde{g}^{t+1}) - \tilde{g}^{t+1}\right\|^2 \mid W^t\right]\right] \\
&\stackrel{(6)}{\leq} (1 - A^M)\mathbb{E}\left[\left\|g^t - \tilde{g}^t\right\|^2\right] + B^M\mathbb{E}\left[\left\|\tilde{g}^{t+1} - \tilde{g}^t\right\|^2\right],
\end{aligned} \tag{33}
$$

Further, we bound the last term in (33). Recall that

$$\tilde{g}^{t+1} = \frac{1}{n} \sum_{i=1}^{n} \tilde{g}_i^{t+1} = \frac{1}{n} \sum_{i=1}^{n} \mathcal{C}_{\tilde{g}_i^t, \nabla f_i(x^t)}^w(\nabla f_i(x^{t+1})). \tag{34}$$

Then,

$$
\begin{aligned}
\mathbb{E}\left[\left\|\tilde{g}^{t+1} - \tilde{g}^t\right\|^2\right] &= \mathbb{E}\left[\left\|\frac{1}{n} \sum_{i=1}^{n} \mathcal{C}_{\tilde{g}_i^t, \nabla f_i(x^t)}^w(\nabla f_i(x^{t+1})) - \tilde{g}_i^t\right\|^2\right] \\
&\leq \frac{1}{n} \sum_{i=1}^{n} \mathbb{E}\left[\left\|\mathcal{C}_{\tilde{g}_i^t, \nabla f_i(x^t)}^w(\nabla f_i(x^{t+1})) - \tilde{g}_i^t\right\|^2\right] \\
&\stackrel{(19)}{\leq} \frac{3}{n} \sum_{i=1}^{n} \mathbb{E}\left[\left\|\mathcal{C}_{\tilde{g}_i^t, \nabla f_i(x^t)}^w(\nabla f_i(x^{t+1})) - \nabla f_i(x^{t+1})\right\|^2\right] \\
&\quad + \frac{3}{n} \sum_{i=1}^{n} \mathbb{E}\left[\left\|\nabla f_i(x^{t+1}) - \nabla f_i(x^t)\right\|^2\right] + \frac{3}{n} \sum_{i=1}^{n} \mathbb{E}\left[\left\|\nabla f_i(x^t) - \tilde{g}_i^t\right\|^2\right] \\
&\stackrel{(6)}{\leq} 3(1 - A^W)\frac{1}{n} \sum_{i=1}^{n} \mathbb{E}\left[\left\|\nabla f_i(x^t) - \tilde{g}_i^t\right\|^2\right] + 3B^W\frac{1}{n} \sum_{i=1}^{n} \mathbb{E}\left[\left\|\nabla f_i(x^{t+1}) - \nabla f_i(x^t)\right\|^2\right] \\
&\quad + \frac{3}{n} \sum_{i=1}^{n} \mathbb{E}\left[\left\|\nabla f_i(x^{t+1}) - \nabla f_i(x^t)\right\|^2\right] + \frac{3}{n} \sum_{i=1}^{n} \mathbb{E}\left[\left\|\nabla f_i(x^t) - \tilde{g}_i^t\right\|^2\right] \\
&\stackrel{\text{Assumption 3}}{\leq} 3(2 - A^W)\mathbb{E}\left[P^t\right] + 3(B^W + 1)L_+^2\mathbb{E}\left[\left\|x^{t+1} - x^t\right\|^2\right] \\
&= 3(2 - A^W)\mathbb{E}\left[P^t\right] + 3(B^W + 1)L_+^2\mathbb{E}\left[R^t\right],
\end{aligned} \tag{35}
$$

where the first inequality follows from Young's inequality. Plugging (35) into (33) we finish the proof:

$$
\begin{aligned}
\mathbb{E}\left[\left\|g^{t+1} - \tilde{g}^{t+1}\right\|^2\right] \leq (1 - A^M)\mathbb{E}&\left[\left\|g^t - \tilde{g}^t\right\|^2\right] + 3B^M(2 - A^W)\mathbb{E}\left[P^t\right] \\
&+ 3B^M(B^W + 1)L_+^2\mathbb{E}\left[R^t\right].
\end{aligned}
$$

$\square$

Having proved the previous lemmas, we can now show the convergence of bidirectional 3PC algorithm.

**Theorem 7.** *Let Assumptions 3 and 5 hold, and let the stepsize in Algorithm 2 be set as*

$$0 \leq \gamma < \left( L_- + L_+ \sqrt{ \frac{6B^M(B^W+1)}{A^M} + \frac{2B^W}{A^M}\left(1 + \frac{3B^M(2-A^W)}{A^M}\right) } \right)^{-1}. \tag{36}$$

*Fix $T$ and let $\hat{x}^T$ be chosen uniformly from $\{x^0, x^1, \cdots, x^{T-1}\}$ uniformly at random. Then*

$$\mathbb{E}\left[ \left\| \nabla f(\hat{x}^T) \right\|^2 \right] \leq \frac{2\Psi^0}{\gamma T}. \tag{37}$$

*where $\Psi^T = f(x^t) - f^{\inf} + \frac{\gamma}{A^M}\left\| g^t - \tilde{g}^t \right\|^2 + \frac{\gamma}{A^W}\left(1 + \frac{3B^M(2-A^W)}{A^M}\right)\frac{1}{n}\sum_{i=1}^n \left\| \tilde{g}_i^t - \nabla f_i(x^t) \right\|^2.$*

*Proof.* We apply Lemma 7 and split the error $\left\| g^t - \nabla f(x^t) \right\|^2$ into two parts

$$f(x^{t+1}) \stackrel{(20)}{\leq} f(x^t) - \frac{\gamma}{2}\left\| \nabla f(x) \right\|^2 - \left( \frac{1}{2\gamma} - \frac{L_-}{2} \right)R^t + \frac{\gamma}{2}\left\| g^t - \nabla f(x^t) \right\|^2$$

$$\stackrel{(18)}{\leq} f(x^t) - \frac{\gamma}{2}\left\| \nabla f(x) \right\|^2 - \left( \frac{1}{2\gamma} - \frac{L_-}{2} \right)R^t + \gamma\left\| \tilde{g}^t - \nabla f(x^t) \right\|^2 + \gamma\left\| \tilde{g}^t - g^t \right\|^2$$

$$\leq f(x^t) - \frac{\gamma}{2}\left\| \nabla f(x) \right\|^2 - \left( \frac{1}{2\gamma} - \frac{L_-}{2} \right)R^t + \frac{\gamma}{n}\sum_{i=1}^n \left\| \tilde{g}_i^t - \nabla f_i(x^t) \right\|^2 + \gamma\left\| \tilde{g}^t - g^t \right\|^2$$

$$= f(x^t) - \frac{\gamma}{2}\left\| \nabla f(x) \right\|^2 - \left( \frac{1}{2\gamma} - \frac{L_-}{2} \right)R^t + \gamma P^t + \gamma\left\| \tilde{g}^t - g^t \right\|^2, \tag{38}$$

where in the last inequality we applied Young's inequality. Subtracting $f^{\inf}$ from both sides of the above inequality, taking expectation and using the notation $\delta^t = f(x^t) - f^{\inf}$, we get

$$\mathbb{E}\left[ \delta^{t+1} \right] \leq \mathbb{E}\left[ \delta^t \right] - \frac{\gamma}{2}\mathbb{E}\left[ \left\| \nabla f(x^t) \right\|^2 \right] - \left( \frac{1}{2\gamma} - \frac{L_-}{2} \right)\mathbb{E}\left[ R^t \right] + \gamma\mathbb{E}\left[ P^t \right] + \gamma\mathbb{E}\left[ \left\| \tilde{g}^t - g^t \right\|^2 \right]. \tag{39}$$

Further, Lemmas 10 and 11 provide the recursive bounds for the last two terms of (39)

$$\mathbb{E}\left[ P^{t+1} \right] \leq (1 - A^W)\mathbb{E}\left[ P^t \right] + B^W L_+^2\mathbb{E}\left[ R^t \right], \tag{40}$$

$$\mathbb{E}\left[ \left\| g^{t+1} - \tilde{g}^{t+1} \right\|^2 \right] \leq (1 - A^M)\mathbb{E}\left[ \left\| g^t - \tilde{g}^t \right\|^2 \right] + 3B^M(2 - A^W)\mathbb{E}\left[ P^t \right]$$
$$+ 3B^M(B^W + 1)L_+^2\mathbb{E}\left[ R^t \right]. \tag{41}$$

Summing up (39) with a $\frac{\gamma}{A^M}$ multiple of (41) we obtain

$$\mathbb{E}\left[ \delta^{t+1} \right] + \frac{\gamma}{A^M}\mathbb{E}\left[ \left\| g^t - \tilde{g}^t \right\|^2 \right] \leq \mathbb{E}\left[ \delta^t \right] - \frac{\gamma}{2}\mathbb{E}\left[ \left\| \nabla f(x^t) \right\|^2 \right] - \left( \frac{1}{2\gamma} - \frac{L_-}{2} \right)\mathbb{E}\left[ R^t \right]$$
$$+ \gamma\mathbb{E}\left[ P^t \right] + \gamma\mathbb{E}\left[ \left\| \tilde{g}^t - g^t \right\|^2 \right]$$
$$+ \frac{\gamma}{A^M}\left( (1 - A^M)\mathbb{E}\left[ \left\| g^t - \tilde{g}^t \right\|^2 \right] \right)$$
$$+ \frac{\gamma}{A^M}\left( 3B^M(2 - A^W)\mathbb{E}\left[ P^t \right] + 3B^M(B^W + 1)L_+^2\mathbb{E}\left[ R^t \right] \right)$$
$$\leq \mathbb{E}\left[ \delta^t \right] - \frac{\gamma}{2}\mathbb{E}\left[ \left\| \nabla f(x^t) \right\|^2 \right] + \frac{\gamma}{A^M}\mathbb{E}\left[ \left\| g^t - \tilde{g}^t \right\|^2 \right]$$
$$- \left( \frac{1}{2\gamma} - \frac{L_-}{2} - \frac{3\gamma B^M(B^W + 1)L_+^2}{A^M} \right)\mathbb{E}\left[ R^t \right]$$
$$+ \gamma\left( 1 + \frac{3B^M(2 - A^W)}{A^M} \right)\mathbb{E}\left[ P^t \right].$$

Then adding the above inequality with a $\frac{\gamma}{A^{\mathrm{W}}}\left(1+\frac{3B^{\mathrm{M}}(2-A^{\mathrm{W}})}{A^{\mathrm{M}}}\right)$ multiple of (40), we get

$$
\begin{aligned}
\mathbb{E}\left[\Psi^{t+1}\right] &= \mathbb{E}\left[\delta^{t+1}\right] + \frac{\gamma}{A^{\mathrm{M}}}\mathbb{E}\left[\left\|g^t-\tilde{g}^t\right\|^2\right] + \frac{\gamma}{A^{\mathrm{W}}}\left(1+\frac{3B^{\mathrm{M}}(2-A^{\mathrm{W}})}{A^{\mathrm{M}}}\right)\mathbb{E}\left[P^{t+1}\right] \\
&\leq \mathbb{E}\left[\delta^t\right] - \frac{\gamma}{2}\mathbb{E}\left[\left\|\nabla f(x^t)\right\|^2\right] + \frac{\gamma}{A^{\mathrm{M}}}\mathbb{E}\left[\left\|g^t-\tilde{g}^t\right\|^2\right] \\
&\quad - \left(\frac{1}{2\gamma} - \frac{L_-}{2} - \frac{3\gamma B^{\mathrm{M}}(B^{\mathrm{W}}+1)L_+^2}{A^{\mathrm{M}}}\right)\mathbb{E}\left[R^t\right] + \gamma\left(1+\frac{3B^{\mathrm{M}}(2-A^{\mathrm{W}})}{A^{\mathrm{M}}}\right)\mathbb{E}\left[P^t\right] \\
&\quad + \frac{\gamma}{A^{\mathrm{W}}}\left(1+\frac{3B^{\mathrm{M}}(2-A^{\mathrm{W}})}{A^{\mathrm{M}}}\right)\left((1-A^{\mathrm{W}})\mathbb{E}\left[P^t\right] + B^{\mathrm{W}}L_+^2\mathbb{E}\left[R^t\right]\right) \\
&\leq \mathbb{E}\left[\delta^t\right] + \frac{\gamma}{A^{\mathrm{M}}}\mathbb{E}\left[\left\|g^t-\tilde{g}^t\right\|^2\right] + \frac{\gamma}{A^{\mathrm{W}}}\left(1+\frac{3B^{\mathrm{M}}(2-A^{\mathrm{W}})}{A^{\mathrm{M}}}\right)\mathbb{E}\left[P^t\right] - \frac{\gamma}{2}\mathbb{E}\left[\left\|\nabla f(x^t)\right\|^2\right] \\
&\quad - \left(\frac{1}{2\gamma} - \frac{L_-}{2} - \frac{3\gamma B^{\mathrm{M}}(B^{\mathrm{W}}+1)L_+^2}{A^{\mathrm{M}}} - \frac{\gamma B^{\mathrm{W}}L_+^2}{A^{\mathrm{W}}}\left(1+\frac{3B^{\mathrm{M}}(2-A^{\mathrm{W}})}{A^{\mathrm{M}}}\right)\right)\mathbb{E}\left[R^t\right]. \quad (42)
\end{aligned}
$$

Thus by Lemma 6 and the choice of the stepsize

$$
0 \leq \gamma < \left(L + L_+\sqrt{\frac{6B^{\mathrm{M}}(B^{\mathrm{W}}+1)}{A^{\mathrm{M}}} + \frac{2B^{\mathrm{W}}}{A^{\mathrm{M}}}\left(1+\frac{3B^{\mathrm{M}}(2-A^{\mathrm{W}})}{A^{\mathrm{M}}}\right)}\right)^{-1}, \quad (43)
$$

the last term in (42) is not positive. By summing up inequalities for $t = 0, 1, \cdots, T-1$, we get

$$
0 \leq \mathbb{E}\left[\Psi^T\right] \leq \Psi^0 - \frac{\gamma}{2}\sum_{i=1}^{T-1}\mathbb{E}\left[\left\|\nabla f(x^t)\right\|^2\right].
$$

Multiplying both sides by $\frac{2}{\gamma T}$ and rearranging we get

$$
\frac{1}{T}\sum_{i=1}^{T-1}\mathbb{E}\left[\left\|\nabla f(x^t)\right\|^2\right] \leq \frac{2\Psi^0}{\gamma T}.
$$

$\square$

## B.4 Convergence for general nonconvex functions

The results in two subsequent subsections set $\mathcal{M}^{\mathrm{W}}$ as a 3PC compressor and $\mathcal{M}^{\mathrm{M}}$ as an indentity one. According to Lemma 2, Adaptive 3PC is a 3PC compressor. Thus, convergence results from (Richtárik et al., 2022) are valid for Adaptive 3PC compressor. It leads us to the following corollary.

**Corollary 3** (Corollary 5.6 of (Richtárik et al., 2022)). *Let Assumptions 2, 3 and 5 hold. Let $\mathcal{M}^W$ and $\mathcal{M}^M$ in Algorithm 1 be* Ada3PC *and identity compressors, respectively, and choose the stepsize $\gamma = \frac{1}{L_-+L_+\sqrt{\frac{B_{\max}}{A_{\min}}}}$. Then, for any $T \geq 1$ we have*

$$
\mathbb{E}\left[\|\nabla f(\hat{x}^T)\|^2\right] \leq \frac{2(f(x^0)-f(x^{\inf}))\left(L_-+L_+\sqrt{\frac{B_{\max}}{A_{\min}}}\right)}{T} + \frac{\mathbb{E}\left[\frac{1}{n}\sum_{i=1}^n\|g_i^0-\nabla f_i(x^0)\|^2\right]}{A_{\min}T}.
$$

*That is, to achieve $\mathbb{E}\left[\|\nabla f(\hat{x}^T)\|^2\right] \leq \varepsilon^2$ for some $\varepsilon > 0$, Algorithm 1 requires*

$$
T = \mathcal{O}\left(\frac{2(f(x^0)-f(x^{\inf}))\left(L_-+L_+\sqrt{\frac{B_{\max}}{A_{\min}}}\right)}{\varepsilon^2} + \frac{\mathbb{E}\left[\frac{1}{n}\sum_{i=1}^n\|g_i^0-\nabla f_i(x^0)\|^2\right]}{A_{\min}\varepsilon^2}\right)
$$

*iterations.*

### B.5 Convergence for PŁnonconvex functions

The setup here is the same as in the previous subsection, except we add the following assumption.

**Assumption 6** (PŁ condition). *Function $f : \mathbb{R}^d \to \mathbb{R}$ satisfies the Polyak-Łojasiewicz (PŁ) condition with parameter $\mu > 0$, i.e.,*

$$\|\nabla f(x)\|^2 \geq 2\mu(f(x) - f^*) \quad \forall x \in \mathbb{R}^d,$$

*where $x^* := \underset{x \in \mathbb{R}^d}{\arg\min} f(x)$ and $f^* := f(x^*)$.*

**Corollary 4** (Corollary 5.9 of (Richtárik et al., 2022)). *Let Assumptions 2, 3, 5 and 6 hold. Let $\mathcal{M}^W$ and $\mathcal{M}^M$ in Algorithm 1 be* Ada3PC *and identity compressors, respectively, and choose the stepsize*

$$\gamma = \min\left\{\frac{1}{L_- + L_+\sqrt{\frac{2B_{\max}}{A_{\min}}}}, \frac{A_{\min}}{2\mu}\right\}.$$

*Then, to achieve $\mathbb{E}\left[f(x^T)\right] - f^* \leq \varepsilon$ for some $\varepsilon > 0$ the method requires*

$$\mathcal{O}\left(\max\left\{\frac{L_- + L_+\sqrt{\frac{B_{\max}}{A_{\min}}}}{\mu}, A_{\min}\right\} \log \frac{f(x^0) - f(x^{\inf}) + \mathbb{E}\left[\frac{1}{n}\sum_{i=1}^n \|g_i^0 - \nabla f_i(x^0)\|^2 \gamma/A_{\min}\right]}{\varepsilon}\right)$$

*iterations.*

## C   Experimental details and extra experiments

All simulations are implemented in Python 3.8 and run on Intel(R) Xeon(R) Gold 6230R CPU cluster with 48 nodes. We fine-tune the stepsize of each considered algorithm with $(2^0, 2^1, \ldots, 2^8)$ multiples of the corresponding theoretical stepsize. As contractive compressor we use Top-$k$ operator. For EF21 and CLAG we use top-1 compressor, which usually the best in practice for these methods. For AdaCGD we choose compressors varying from full compression (skip communication) to zero compression of features (sending full gradient). In order to provide fair comparisons, we choose master compressor $\mathcal{M}^M$ as identity operator in these experiments. For the stopping criterion we choose communication cost of the algorithm.

We use the setup described in Richtárik et al. (2022), namely logistic regression with nonconvex regularizer:

$$\min_{x \in \mathbb{R}^d} \left[ f(x) := \frac{1}{N} \sum_{i=1}^{N} \log(1 + e^{-y_i a_i^\top x}) + \lambda \sum_{j=1}^{d} \frac{x_j^2}{1+x_j^2} \right],$$

where $a_i \in \mathbb{R}^d$, $y_i \in \{-1, 1\}$ are the training samples and labels with regularization hyperparameter $\lambda > 0$ chosen at $\lambda = 0.1$ level. We solve this problem using LIBSVM Chang & Lin (2011) datasets *phishing, a1a, a9a*. Each dataset has been evenly split into $n = 20$ equal parts where each part represents a separate client. Figures 3-6 compare AdaCGD with LAG, EF21 and their generalization CLAG. Figure 4 provides the comparison in convex regime, *i.e.* $\lambda = 0$, for phishing dataset. In the experiments, AdaCGD is shown to be comparable and in some cases superior to CLAG and always superior to LAG. In other words, AdaCGD efficiently complements CLAG and other 3PC methods.

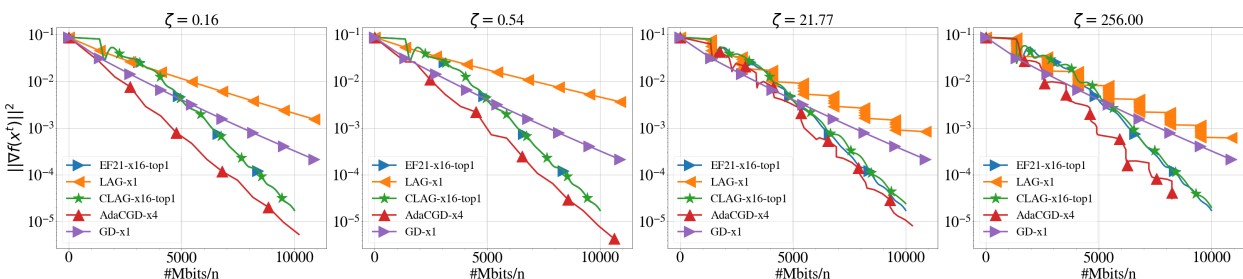

Figure 3:  Comparison of LAG, CLAG, EF21 and GD with AdaCGD on *phishing* dataset.

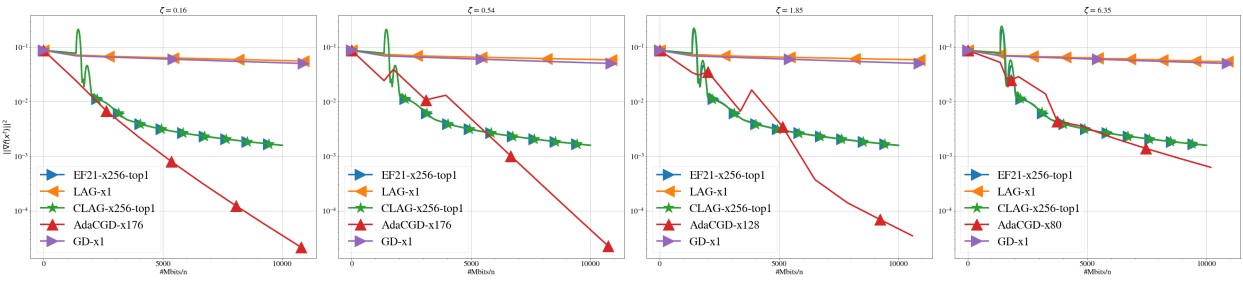

Figure 4:  Comparison of LAG, CLAG, EF21 and GD with AdaCGD on *phishing* dataset in the convex regime *i.e.* $\lambda = 0$.

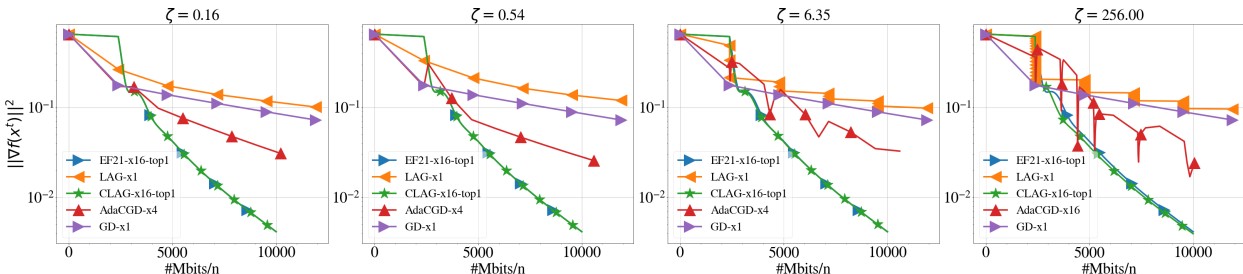

Figure 5: Comparison of LAG, CLAG, EF21 and GD with AdaCGD on *a1a* dataset.

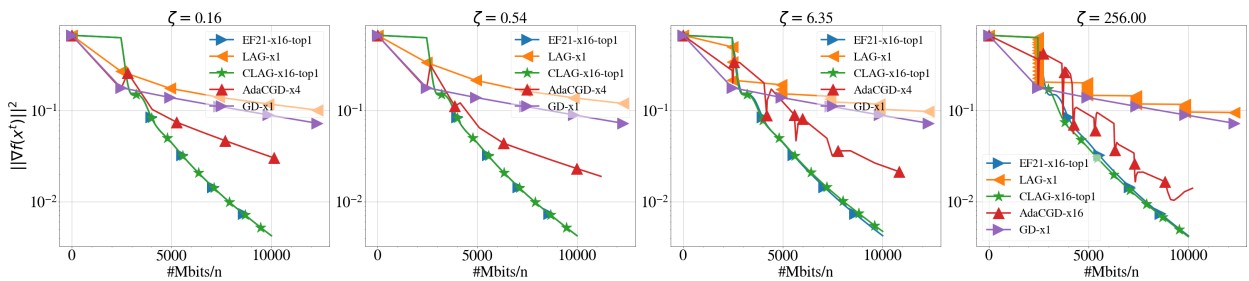

Figure 6: Comparison of LAG, CLAG, EF21 and GD with AdaCGD on *a9a* dataset.

# D   Limitations

The main limitations of the work are assumptions we make upon functions $f_i$ of the problem 1. But, on the other hand, these assumptions govern the convergence rates we report: for example, we cannot show linear rate for convex functions due to the fundamental lower bound (Nesterov et al., 2018).

Another limitation comes from the analysis of Bidirectional `3PC` algorithm (Theorem 7). We show the analysis only for general nonconvex functions.

