# OpenReview forum: "Adaptive Compression for Communication-Efficient Distributed Training"
_TMLR — Accepted by TMLR_

### Review · Reviewer_NXKS · 2023-01-17

**Summary Of Contributions:**

The authors have proposed a method that adaptively updates the compression scheme and levels during the optimization process, extended their consideration to bidirectional compression, and provided convergence  guarantees in a number of standard settings. They focus on a biased and LAG-based compression and use the estimated $\alpha_j$'s in 3PC compression to switch among $m$ EF21-based compression outputs. The theory is based on PL and smoothness assumptions, which is standard in optimization.

**Audience:**

Yes

**Claims And Evidence:**

No

**Requested Changes:**

I do not think this paper is ready for publication mainly due to significant lack of comparison with related work. The authors should properly (algorithmically, theoretically, and empirically) compare with the previous work that proposed adaptive quantization levels and sparsification (detailed comment is provided above). Also the motivation behind bidirectional compression, additional tuning effort, overestimation of  $\alpha_j$'s in Adaptive 3PC compressor, its mean squared analysis, and difficulty of convex and smooth analysis in the considered setting compared to similar analyses for LAG and 3PC should be carefully addressed.




**Strengths And Weaknesses:**

I think it is interesting to consider adaptive variants of biased compression schemes. The paper is relatively well-written. However, this paper has significant issues and requires substantial work. I do not think the paper is ready for publication.

-----

My major concern is that several closely related work have not been cited/compared (Zhang et al., 2017; Wen et al., 2017; Wang et al., 2019; Guo et al., 2020; Faghri et al., 2020; Zhong et al, 2021; Agrawal et al., 2021; Alimohammadi et al., 2022). This is really unfortunate that all those work are ignored. Adaptive compression has been studied extensively in various scientific/engineering areas. In distributed ML, adapting the communication frequency in local SGD, adapting the number of quantization levels, and adaptive quantization levels/sparsification have been studied extensively (Wang et al., 2019; Guo et al., 2020; Faghri et al., 2020;  Agrawal et al., 2021; Alimohammadi et al., 2022). In particular, the authors should properly (algorithmically, theoretically, and empirically) compare with the previous work that proposed adaptive quantization levels and sparsification .


Hantian Zhang, Jerry Li, Kaan Kara, Dan Alistarh, Ji Liu, and Ce Zhang. ZipML: Training linear models with end-to-end low precision, and a little bit of deep learning. In International Conference on Machine Learning (ICML), 2017.

Wei Wen, Cong Xu, Feng Yan, Chunpeng Wu, Yandan Wang, Yiran Chen, and Hai Li. TernGrad: Ternary gradients to reduce communication in distributed deep learning. In Advances in Neural Information Processing Systems (NeurIPS), 2017.

Jianyu Wang and Gauri Joshi. Adaptive communication strategies to achieve the best error- runtime trade-off in local-update SGD. In Proceedings of Machine Learning and Systems (MLSys), 2019.

Jinrong Guo, Wantao Liu, Wang Wang, Jizhong Han, Ruixuan Li, Yijun Lu, and Songlin Hu. Accelerating distributed deep learning by adaptive gradient quantization. In IEEE International Conference on Acoustics, Speech and Signal Processing (ICASSP), 2020.

Fartash Faghri, Iman Tabrizian, Ilia Markov, Dan Alistarh, Daniel M. Roy, and Ali Ramezani-Kebrya. Adaptive gradient quantization for data-parallel SGD. In Advances in Neural Information Processing Systems (NeurIPS), 2020.

Yuchen Zhong, Cong Xie, Shuai Zheng, and Haibin Lin. Compressed communication for distributed
training: Adaptive methods and system. arXiv preprint arXiv:2105.07829, 2021.

Saurabh Agarwal, Hongyi Wang, Kangwook Lee, Shivaram Venkataraman, and Dimitris Papailiopoulos. Adaptive gradient communication via critical learning regime identification. In Proceedings of Machine Learning and Systems (MLSys), 2021.

Mohammadreza  Alimohammadi, Ilia Markov, Elias Frantar, and Dan Alistarh. L-GreCo: An efficient and general framework for layerwise-adaptive gradient compression. arXiv preprint arXiv:2210.17357, 2022.

-----

The motivation of considering ${\cal M}^M$ at the server is not very clear. The downlink capacity in communication network is typically significantly more than the uplink. Also the server needs to broadcast only a single vector to all workers. So the additional complexity of compressing at the worker side given much smaller communication load and higher downlink capacity is not clear.

-----

Tuning this LAG-based method is significantly more challenging than unbiased and compressed variants of SGD mainly due to additional hyperparameters like $\zeta$ and also the bias in compression. I am not sure whether the proposed scheme can be used in practical distributed settings.

-----

In the definition of 3PC compressors, $h$, $x$, and $y$ are treated as independent quantities while they are indeed statistically correlated considering e.g., (5). I think A and B obtained from Definition 3 are quite conservative (overestimated) and the corresponding bounds are loose.  The same comment applies to the overestimation of $\alpha_j$'s in Adaptive 3PC compressor (Section 4.2). Such overestimation will lead to highly suboptimal (in terms of mean squared compression error) but computationally expensive compressor.

-----

The analysis based on PL and smoothness assumptions is quite standard. Not sure the emphasis on nonconvex analysis is really needed since such rates are indeed expected. I am curious to understand the additional difficulty of convex and smooth analysis like Theorem 6 compared to similar analyses for LAG and 3PC (simple compression only at workers).
Some results like Lemma 2 are trivial maybe it is better to put the proof to the appendix.

-----

The author should improve presentation of the paper e.g., the paragraph following Definition 1; the notation of compression operator in (2) is different from that of Algorithm 1.

---

> ### Author Response · Authors · 2023-04-05
> **Response to Reviewer NXKS (Part 1)**
>
> > My major concern is that several closely related work have not been cited/compared
>
> We thank the reviewer for their diligent review of our manuscript and feedback. However, **we could not agree with the statement that our paper has a significant lack of comparison with related work**. We devoted the entire section (Motivation and Background) of the paper to discussing and comparing three different classes of SOTA methods with more than 40 references. We summarized the comparisons with the closest works in Tables 1 and 2, where we highlighted both the analysis and method's differences. Upon careful examination of Referee's response, we have included references to the closest mentioned works of Zhong et al. and Alimohammadi et al., and added the comparisons with their methods to Table 1. The work of Alimohammadi et al. is a **very recent work** was released after the completion of this work, and we appreciate the Referee bringing it to our attention. **Comparing to this work, our analysis provides better theoretical convergence rates, under more standard assumptions on the compressor types (see footnote 6 on Table 1.)**. The work of Zhong et al. uses **constant compressors** and provides adaptivity in the stepsize, which is **a very different approach from our's**. We also added the comparison with the work of Faghri et. al. to Table 1. Even though this work **deals with quantizer compressors** only, it provides explicit theoretical convergence rates which we could use for comparison. As can be seen from Table 1, **our method outperforms the existing approaches** in terms of convergence rates in non-convex settings and our analysis covers a broader range of settings, including convex, strongly convex, non-convex, functions with analytical P\L properties and support for compression in both directions (server and client).
>
> However, **we do not consider the other works mentioned by the reviewer as closely related to our work** for several reasons. Firstly, some ot these works (Zhong et al.,  Zhang et. al., Wang et al, Wen et. al.) **uses conceptually different classes of methods**, which could not be closely compared without breaking the narrative. As mentioned above, the work of Zhong et al. uses **constant compressors and provides adaptivity in the stepsize, which is a very different approach, that rather belongs to a group of methods with adaptive stepsizes not adaptive compression**. The work of Zhang et. al. **is not even related to a distributed setting**, so we do not see how the comparison with this work would improve our manuscript. Furthermore, Zhang et al. suggest selecting quantization levels in an intelligent manner, as a component of preprocessing. In other words, once established at the outset of the optimization process, **the quantization levels remain fixed and do not undergo any subsequent changes**. Wang et. al. focus on **different aspects** of distributed machine learning, such as **adaptive communication frequency in local SGD, which does not address the problem of adaptive compression** and is more related to the pure LAG context, which we extensively discussed and compared in both theory (Tables 1, 2) and experiments (Figure 1). Secondly, **most of these works (Agarwal et al., Wen et. al., Guo et. al.) are experimental** and **do not provide a comprehensive theoretical analysis** of their proposed methods (Agarwal et al., Wen et al.), with **clear convergence guarantees and rates** (Guo et al.). In contrast, **our work's main focus is** on the development of **novel adaptive compression methods** for distributive learning **with strong theoretical guarantees**. Wen et al. have introduced a novel quantization **approach known as TernGrad**, which **is non-adaptive in nature**. Specifically, given an input vector $v$, TernGrad maps it to another vector $Q(v)$, where the output is determined solely by the input vector $v$. In contrast, our proposed method, AdaCGD, incorporates information about the gradient $\nabla f(x^t)$, in addition to the input vector, in order to create an adaptive compressor that is responsive to the evolving iterates $x^t$. Guo et al. also deal with quantizer compressors only and does not provide explicit convergence analysis for their methods. **All these works are not closely related to our method** to provide a more detailed comparison; nevertheless, **we followed the reviewer's suggestion and referenced these works** in the Motivation and Background section to solidify the claims and improve the accessibility of the paper.

---

> ### Author Response · Authors · 2023-04-05
> **Response to Reviewer NXKS (Part 2)**
>
> In summary, in response to the reviewer's claim regarding lack of comparison, **we have revised the manuscript**, adding a few works (Zhong et al., Alimohammadi et al., Faghri et. al.) to the comparison table and citations to a few other papers in the Motivation and Background section. **These revisions are minor and do not modify any statement in the paper**. Compared to **all of these works, our method is highly novel**, and **our analysis is more comprehensive**, providing **superior theoretical convergence rates** (as illustrated in Tables 1 \& 2). Therefore, **we respectfully disagree** with the reviewer's assertion that our manuscript lacks comparisons with the mentioned works.We hope **our modifications to the paper and this discussion illustrate this**.

---

> ### Author Response · Authors · 2023-04-05
> **Response to Reviewer NXKS (Part 3)**
>
> > The motivation of considering at the server is not very clear.
>
> The existing literature on communication compression in distributed optimization addresses two communication regimes: asymmetric and symmetric. The asymmetric regime assumes that the cost of downlink communication is negligible, and most papers consider this assumption. While this regime models situations where the uplink and downlink speeds have a significant disparity, it is only valid to a certain extent. At some point the uplink compression level is so high that the uplink speed per communication round matches the fast downlink speed per communication round. Beyond this compression level, it would be inaccurate to not account for downlink communication. For large-scale machine learning applications, where the data dimensionality can be significant this point can be reached quickly. In such cases, compressing downlink communication is necessary for further improvement.
>
> Another regime is the much more challenging for analysis symmetric regime, in which the cost of downlink communication is not assumed to be negligible. This regime is largely overlooked by the community because existing methods which support bidirectional compression suffer from a multiplicative dependence on the variances that come from the uplink and downlink compressors, making these methods less effective and interesting from a practical and theoretical point of view. So in this regime, compression in both directions are important and could lead to a better convergence.
>
> > Tuning this LAG-based method is significantly more challenging than unbiased and compressed variants of SGD mainly due to additional hyperparameters
>
> We appreciate the reviewer's feedback regarding the hyperparameters of our proposed adaptive compression method. However, we argue that the additional hyperparameters introduced in our method are minimal. The proposed adaptive method, apart from the specified compressors, only uses one additional hyperparameter, the threshold parameter $\zeta$. It can be set once based on empirical evaluation, and its value does not change during the training process. Moreover, the bias in compression is not a significant issue in practical settings since our proposed method employs contractive compressors that have been widely used in the literature and are known to offer better empirical convergence rates than unbiased compressors.
>
> The main advantage of our method is that it encompasses existing practical methods such as EF21, LAG, and CLAG as its partial cases, providing an additional degree of freedom for designers to improve existing methods with already tuned hyperparameters. Concerning practical usability, our proposed method can be employed in practical distributed settings to reduce overall communication load and training time by adaptively varying the compression level for each client based on their local data. Our method is compatible with the existing communication infrastructure and requires minimal modifications. We also provide experimental evaluation in our paper that demonstrates the effectiveness of our method in some distributed settings.
>
> > In the definition of 3PC compressors variables are treated as independent quantities while they are indeed statistically correlated
>
> The dependence on $x$, $y$, and $h$ in our analysis is not an issue at all, as it is a standard practice in optimization theory. The reviewer's criticism is not specific to our paper but rather a general property of many optimization analyses that require a universal bound on some quantity. In our case, we need to establish an inequality that holds for all possible values of $x$, $y$, and $h$ to prove our main result. However, the proof does not require to consider all possible combinations of these values explicitly.

---

> ### Author Response · Authors · 2023-04-05
> **Response to Reviewer NXKS (Part 4)**
>
> >  The analysis based on PL and smoothness assumptions is quite standard. Not sure the emphasis on nonconvex analysis is really needed since such rates are indeed expected. I am curious to understand the additional difficulty of convex and smooth analysis like Theorem 6 compared to similar analyses for LAG and 3PC (simple compression only at workers). Some results like Lemma 2 are trivial maybe it is better to put the proof to the appendix.
>
> Our paper's main contribution is a novel universal method for distributed optimization that achieves SOTA communication complexity in the challenging symmetric communication regime, as well as a theoretical analysis that rigorously justifies our method's superior performance. While it is true that we provide complete theoretical analysis in various regimes including the analysis based on standard PL and smoothness assumptions,we do not consider it as a drawback. Instead, we believe it is a fair and novel theoretical benchmark under conventional and realistic assumptions. Moreover, the analysis of our proposed method in other regimes, like convex and bi-directional, including Theorem 6, is novel result for the whole family of error-feedback methods and substantially more complex and challenging than that of EF21 3PC, both of which only consider simple compression at the workers. Regarding the Lemma 2 simplicity, we believe that this Lemma is an important building block in our analysis, and its proof provides important insights into the properties of the adaptive 3PC compressor. Therefore, we believe that including the proof in the main text is justified.
>
> > he author should improve presentation of the paper e.g., the paragraph following Definition 1; the notation of compression operator in (2) is different from that of Algorithm 1.
>
> We thank reviewer for the suggestion. We unified the notation of compression operator in Algorithm 1

---

### Review · Reviewer_xFzN · 2023-01-20

**Summary Of Contributions:**

The paper proposes an adaptive variant of the 3-point compression [Richtarik 2022]. The proposed adaptive scheme is essentially an extension of the LAG mechanism to m 3PC compressors accompanied with corresponding condition functions as follows: the adaptive 3PC sequentially checks the conditions and uses the first 3PC compressor whose corresponding condition function is satisfied. For the case of AdaCGD, the condition function can be interpreted as a measure of compression error.
The authors provided a new theoretical analysis for the convergence rate of 3PC, as well as bidirectional 3PC (i.e., both workers and server use compresses their data). Since adaptive 3PC can be interpreted as a more general 3PC, the convergence results of 3PC are readily extended to the adaptive 3PC and AdaCGD. Finally, they used a simple logistic regression problem over LIBSVM datasets to compare the proposed algorithm with EF21, LAG, CLAG, and GD.

**Audience:**

Yes

**Broader Impact Concerns:**

there is no concern on the ethical implications of the work.

**Claims And Evidence:**

Yes

**Requested Changes:**

1. How the theoretical analysis of the paper for 3PC compares w.r.t, [Richtarik, 2022], e.g., theorems 5.5 and 5.8
2. The experimental results actually show that AdaCGD is worse than CLAG and EF21 on *a1a* and *a9a* datasets, and better than *phishing* dataset by a relatively small amount. Hence, the extra computations by AdaCGD does not necessarily translate to better convergence rate or fewer communication bits.
3. How that $\zeta$ affects the algorithm? Intuitively, larger values make AdaCGD behave like CLAG. Does this actually happens in practice, i.e., AdaCGD mostly uses the first 3PC compressor?
4. How the algorithms compare in the convex setting, e.g., setting $\lambda=0$ in the experiments.
5. It would be better to give the experiments' details in the main part, not the appendix, e.g.,
	- what is GD-x1 algorithm. (not mentioned even in the appendix)
	- what does '-xN' stands for in the legend of the figures.

**Strengths And Weaknesses:**

**Strengths**:

The adaptivity mechanism and the proposed algorithm are clearly explained and the theoretical analysis seems correct. Moreover, the results and contributions are clearly compared against some of the related works (tables 1 & 2), although there are still unclear points.

**Weaknesses**:
- There is a disconnect between the theoretical analysis and the proposed adaptive 3PC method. The theoretical results are mainly derived for the 3PC under almost the same assumptions as in [Richtarik 2022], hence can be seen as an extended analysis to that paper. Then the authors show that the adaptive 3PC is indeed a 3PC compressor with $A_{min}$ and $B_{max}$ parameters and the convergence results boil down to the 3PC results with $A_{min}$ and $B_{max}$ replacing the original A and B parameters. Although, I assume that the actual analysis of the adaptive 3PC would be much more involved and possibly complex, but at least the authors should devote some discussions to how the adaptiveness actually affects the convergence.
- The other issue is the simulations. Some details are missing, the setup is very simplistic and the results actually doesn't show AdaCGD is superior to e.g., CLAG.

(minor issue, not affecting the evaluation) It is worth noting that the AdaCGD requires the actual gradient of the functions, hence it is not clear whether in more practical scenarios where only an estimate of the gradient is available, the proposed algorithm works or have comparable convergence rate.

---

> ### Author Response · Authors · 2023-04-05
> **Response to Reviewer's (xFzN) Suggestions Regarding a Disconnect Between the Theoretical Analysis and Adaptive 3PC Method**
>
> >The adaptivity mechanism and the proposed algorithm are clearly explained and the theoretical analysis seems correct. Moreover, the results and contributions are clearly compared against some of the related works (tables 1 \& 2), although there are still unclear points.
>
> >Weaknesses:
> >There is a disconnect between the theoretical analysis and the proposed adaptive 3PC method. The theoretical results are mainly derived for the 3PC under almost the same assumptions as in [Richtarik 2022], hence can be seen as an extended analysis to that paper. Then the authors show that the adaptive 3PC is indeed a 3PC compressor with and parameters and the convergence results boil down to the 3PC results with and replacing the original A and B parameters. Although, I assume that the actual analysis of the adaptive 3PC would be much more involved and possibly complex, but at least the authors should devote some discussions to how the adaptiveness actually affects the convergence.
>
> We appreciate the reviewer's thoughtful feedback regarding our paper and would like to provide additional clarification on the theoretical analysis and adaptiveness of the proposed Ada3PC algorithm. Our central idea in the paper was to develop an algorithm that auto-tunes to the gradient estimate gap, allowing for compression of the information sent to be varied depending on this gap (see added Fig. 2).  Intuitively, when the gradients on the neighboring time steps have a smaller gap, it leads to smaller updates, allowing us to apply larger compression without losing information. Conversely, when there is a more significant gap between gradients, it means more informative updates that are more susceptible to higher compression, hence requiring less compression. This intuition forms the foundation of the proposed adaptive cost functions based on the Ada3PC algorithm, enabling more efficient compression of new gradients and faster convergence to the solution. The compressor distributions during training on the phishing dataset are illustrated in Figure 2 (added in revision), showing that there is a regime that favors non-extreme compressors with a specific hyperparameter $\zeta$, leading to better experimental results compared to other 3PC approaches (see Fig. 1 and Fig. 2 of the main text). We followed the reviewer's suggestion and enlarge the experiment's section with Fig. 2 and discussion.
>
> We agree with the Referee's opinion that a detailed analysis of adaptivity can potentially result in better rates. Nonetheless, the challenge lies in the unavailability of techniques to analyze algorithms with heuristic update rules, such as the LAG condition. Consequently, it is difficult to estimate, under practical assumptions, the frequency of trigger condition activation and anticipate the selected compression. This is a limitation not only of our paper, but also an unresolved challenge for the entire research community.
>
> Besides this, we want to reiterate that in this work we provide a new analysis with sharp convergence bounds for Ada3PC/3PC family of methods in the strongly convex, convex, and nonconvex settings. Analysis in convex setting is the new theoretical SOTA for entire error-feedback family of methods. We also present entirely new theoretical analysis for bidirectional compression of Ada3PC/3PC framework. We appreciate your feedback and believe that these additions will significantly improve the clarity of our paper. We incorporate this discussion to the experimental section of the manuscript.
>
> >The other issue is the simulations. Some details are missing, the setup is very simplistic and the results actually doesn't show AdaCGD is superior to e.g., CLAG.
>
> The primary objective of this work was developing a novel method for distributed learning with adaptive compression with strong and comprehensive theoretical convergence guarantees. The main purpose of our experiments was to verify the predictive capability of our theory. While the results may not show a clear superiority of AdaCGD over CLAG in terms of convergence rate on all datasets, we believe that they still support the claim that our proposed method is effective and can achieve competitive or better in some cases performances with other state-of-the-art methods while providing more flexibility and adaptivity. We have taken into consideration the suggestions provided by the reviewer and revise the experimental section providing more details.

---

> ### Author Response · Authors · 2023-04-05
> **Response to Reviewer's (xFzN) Requested Changes**
>
> > How the theoretical analysis of the paper for 3PC compares w.r.t, [Richtarik, 2022], e.g., theorems 5.5 and 5.8
>
> The results of comparison are summarized in the Table 2, where we provide the comparison of convergence guarantees results of methods employing lazy aggregation. In this paper, we provide the novel theoretical analysis for these new class of adaptive compressors with bi-directional compression as well as new SOTA convergence analysis in convex regime.
>
> > How that affects the algorithm? Intuitively, larger values make AdaCGD behave like CLAG. Does this actually happens in practice, i.e., AdaCGD mostly uses the first 3PC compressor?
>
> Yes, by controlling $\zeta$ parameter we can switch AdaCGD between different regimes. As expected, when $\zeta=0$, AdaCGD is equivalent to EF21, which tends to choose the maximum compression level across different compressors. Conversely, with a larger $\zeta$ value, the AdaCGD behaves more like LAG/CLAG and has an increased number of skip connections. However, there is a range of $\zeta$ values (0.16, 0.54, 1.85, and 6.35) where the algorithm benefits from adaptivity and utilizes the full spectrum of compressors. We follow the reviewer's suggestion and add additional figure (fig. 2) and discussion to the main text, describing the use of compressors during the training with different $\zeta$ parameters.
>
> > The experimental results actually show that AdaCGD is worse than CLAG and EF21 on a1a and a9a datasets, and better than phishing dataset by a relatively small amount. Hence, the extra computations by AdaCGD does not necessarily translate to better convergence rate or fewer communication bits.
>
> While the results may not show a clear superiority of AdaCGD over CLAG in terms of convergence rate on all datasets, we believe that they still support the claim that our proposed method is effective and can achieve competitive or better in some cases performances with other state-of-the-art methods while providing more flexibility and adaptivity.
>
> > How the algorithms compare in the convex setting, e.g., $\lambda=0$ setting in the experiments.
>
>  We followed the reviewer's suggestion and add additional experiments in the convex setting in the appendix (Figure 4). The algorithm's behavior in this setting is similar to the non-convex case, i.e., **AdaCGD** converges faster than **CLAG**,  **LAG**, **GD** and **EF21**.
>
> > It would be better to give the experiments' details in the main part, not the appendix, e.g., what is GD-x1 algorithm. (not mentioned even in the appendix) what does '-xN' stands for in the legend of the figures.
>
> We clarify this in the figures caption (-x1 GD stands for Gradient Decent with theoretical stepsize multiplied by 1) and we have taken into consideration the suggestions provided by the reviewer and revised the experimental section. As a result, we have included additional details, a new figure, and more comprehensive captions for the figures.

---

### Review · Reviewer_bcXK · 2023-03-23

**Summary Of Contributions:**

This work proposes an Adaptive Three Point Compressor (Ada3PC) framework, which is a variant of the 3PC framework proposed in (Richtárik et al., 2022) that supports adaptive or tunable compression level. The authors provided unified analysis based on the 3PC framework and established SOTA convergence guarantees for convex, strongly convex and non-convex objectives, extending the existing theoretical development of the 3PC framework. The new Ada3PC induces the AdaCGD method, which covers several existing methods such as EF21, CLAG, LAG as specially cases, and extends their convergence analysis in the convex case. Numerical results demonstrated the benefits of having adaptive compression level.

**Audience:**

Yes

**Broader Impact Concerns:**

N/A, a mainly theoretic work

**Claims And Evidence:**

Yes

**Requested Changes:**

Add some discussion of the convex analysis of the bidirectional version of AdaCGD. If the convex cases can also be covered here, the whole picture will be more complete and Table 2 will be better supported.

**Strengths And Weaknesses:**

Strengths:
- A very well-written paper with a nice flow of ideas. The background and existing literature are discussed in very details.
- Unified theoretical analysis framework with SOTA convergence results.
- Nice extensions of the 3PC framework to increase the potential practical implication of such theoretic framework.

Weaknesses:
- I wonder why the bidirectional method is only analyzed for the non-convex case, is there any specific difficulty?

---

### Decision · Action_Editors · 2023-06-08

**Recommendation:** Accept as is

**Comment:**

The manuscript can be accepted as-is, but I ask the authors to bring back some of the results from appendix to main body (as suggested by Reviewer xFzN) or, at the very least, clearly mention that there exists datasets where Ada3PC is clearly outperformed by other baselines.

**Audience:**

This is a fairly specialized, one might almost say incremental, contribution that will be of interest to theoreticians and practitioners of federated learning approaches.

**Claims And Evidence:**

This paper proposes a new algorithm for federated learning that aims at reducing communication cost of gradient updates using compression. More precisely, the algorithm proposes Ada3PC to compress with an adaptive compression strength, those gradients (as in Lines 6, 10 of Alg. 1). The authors analyze that framework, exhibit favorable rates compared to baseline methods, and illustrate performance in experiments, comparing with 4 baselines (2 of which are particular cases of their framework for a specific choice of hyperparameters). Reviewers have expressed a borderline accept call overall, with 2 lean accept and 1 reject. Although I share some of the initial concerns of reviewer NXKS w.r.t. presentation / novelty / benchmarking, I believe the authors have done a significant step in clarifying what they believe is a reasonable space of competitors. Following the two other reviewers, I agree that the paper is well written, provides an exhaustive motivation section that is interesting for a general audience. I therefore suggest to accept.


minor comments:
"In the experiments,AdaCGDis shown to be comparable and in some cases superior to CLAG". You should rather say: also say more clearly inferior in some cases, as mentioned by xFzN with whom I agree.